# Stochastic Primal-Dual Method for Empirical Risk Minimization with $\mathcal{O}(1)$ Per-Iteration Complexity

**Conghui Tan**[*]
The Chinese University of Hong Kong
chtan@se.cuhk.edu.hk

**Tong Zhang**
Tencent AI Lab
tongzhang@tongzhang-ml.org

**Shiqian Ma**
University of California, Davis
sqma@math.ucdavis.edu

**Ji Liu**
Tencent AI Lab, University of Rochester
ji.liu.uwisc@gmail.com

## Abstract

Regularized empirical risk minimization problem with linear predictor appears frequently in machine learning. In this paper, we propose a new stochastic primal-dual method to solve this class of problems. Different from existing methods, our proposed methods only require $\mathcal{O}(1)$ operations in each iteration. We also develop a variance-reduction variant of the algorithm that converges linearly. Numerical experiments suggest that our methods are faster than existing ones such as proximal SGD, SVRG and SAGA on high-dimensional problems.

## 1 Introduction

In this paper, we consider the convex regularized empirical risk minimization with linear predictors:

$$\min_{x \in X} \left\{ P(x) \triangleq \frac{1}{n} \sum_{i=1}^{n} \phi_i(a_i^\top x) + g(x) \right\}, \tag{1}$$

where $X \subset \mathbb{R}^d$ is a convex closed feasible set, $a_i \in \mathbb{R}^d$ is the $i$-th data sample, $\phi_i$ is its corresponding convex closed loss function, and $g(x) : X \to \mathbb{R}$ is a convex closed regularizer for model parameter $x$. Here we assume the feasible set $X$ and the regularizer function $g(x)$ are both separable, i.e.,

$$X = X_1 \times \cdots \times X_d \quad \text{and} \quad g(x) = \sum_{j=1}^{d} g_j(x_j). \tag{2}$$

Problem (1) with structure (2) generalizes many well-known classification and regression problems. For example, support vector machine is with this form by choosing $\phi_i(u) = \max\{0, 1 - b_i u\}$ and $g(x) = \frac{\lambda}{2} \|x\|_2^2$. Other examples include $\ell_1$ logistic regression, $\ell_2$ logistic regression, and LASSO. One popular choice for solving (1) is the proximal stochastic gradient descent method (PSGD). In each iteration of PSGD, an index $i$ is randomly sampled from $\{1, 2, \ldots, n\}$, and then the iterates are updated using only the information of $a_i$ and $\phi_i$. As a result, the per-iteration cost of PSGD is $\mathcal{O}(d)$ and independent of $n$.

It is well known that PSGD converges at a sub-linear rate [9] even for strongly convex problems, due to non-diminishing variance of the stochastic gradients. One line of research tried is dedicated to improve the convergence rate of PSGD by utilizing the finite sum structure in (1). Some representative

---

[*]This work was done while Conghui Tan was a research intern at Tencent AI lab.

works include SVRG [5, 16], SDCA [13, 14], SAGA [4] and SPDC [18]. All these accelerated variants enjoy linear convergence when $g(x)$ is strongly convex and all $\phi_i$'s are smooth.

Since all of these algorithms need to sample at least one data $a_i$ in each iteration (so their per-iteration cost is at least $O(d)$), their potential drawbacks include: 1) they are not suitable for the distributed learning with features distributed; 2) it may incur heavy computation per iteration in the high-dimensional case, i.e., when $d$ is very large.

**Our contributions.** In this paper, we explore the possibility of accelerating PSGD by making each iteration more light-weighted: only one coordinate of one data is sampled, i.e., one entry $a_{ij}$, in each iteration of the algorithm. This leads to a new algorithm, named SPD1 (stochastic primal-dual method with $\mathcal{O}(1)$ per-iteration complexity), whose per-iteration cost is only $\mathcal{O}(1)$. We prove that the convergence rate of the new method is $\mathcal{O}(1/\sqrt{t})$ for convex problems and $\mathcal{O}(\ln t/t)$ for strongly convex and smooth problems, where $t$ is the iteration counter. Moreover, the overall computational cost is the same as PSGD in high-dimensional settings. Therefore, we managed to reduce the per-iteration complexity from $\mathcal{O}(d)$ to $\mathcal{O}(1)$ while keep the total computational cost at the same order. Furthermore, by incorporating the variance reduction technique, we develop a variant of SPD1, named SPD1-VR, that converges linearly for strongly convex problems. Comparing with existing methods, our SPD1 and SPD1-VR algorithms are more suitable for distributed systems by allowing the flexibility of either feature distributed or data distributed. An additional advantage of our $\mathcal{O}(1)$ per-iteration complexity algorithms is that they are more favorable by asynchronous parallelization and bring more speedup since they admit much better tolerance to the staleness caused by asynchronity. Our numerical tests indicate that our methods are faster than both PSGD and SVRG on high-dimensional problems, even in single-machine setting.

We notice that [6] and [17] used similar ideas in the sense that in each iteration only one coordinate of the iterate is updated using one sampled data. However, we need to point out that their algorithms still need to sample the full vector $a_i$ to compute the directional gradient, and thus the per-iteration cost is still $\mathcal{O}(d)$.

## 1.1 Notation

We use $A \in \mathbb{R}^{n \times d}$ to denote the data matrix, whose rows are denoted by $a_i$, $i = 1, 2, \ldots, n$. We use $a_{ij}$ to denote the $j$-th entry of $a_i$. $[n]$ denotes the set $\{1, 2, \ldots, n\}$. $x^t$ denotes the iterate in the $t$-th iteration and $x_j^t$ is its $j$-th entry. We always use $\|w\|$ to denote the $\ell_2$ norm of $w$ unless otherwise specified.

For function $f(z)$ whose domain is $Z$, its proximal mapping is defined as

$$\text{prox}_f(z) = \underset{z' \in Z}{\arg\min} \left\{ f(z') + \frac{1}{2}\|z' - z\|^2 \right\}. \tag{3}$$

We use $\partial f(z)$ to denote the subdifferential of $f$ at point $z$. $f$ is said to be $\mu$-strongly convex $(\mu > 0)$ if

$$f(z') \geq f(z) + s^\top (z' - z) + \frac{\mu}{2}\|z' - z\|^2 \quad \forall s \in \partial f(z), \forall z, z' \in Z.$$

The conjugate function of $\phi_i : \mathbb{R} \to \mathbb{R}$ is defined as

$$\phi_i^*(y) = \sup_{x \in \mathbb{R}} \left\{ y \cdot x - \phi_i(x) \right\}. \tag{4}$$

Function $\phi_i$ is $L$-Lipschitz continuous if

$$|\phi_i(x) - \phi_i(y)| \leq L|x - y|, \quad \forall x, y \in \mathbb{R},$$

which is equivalent to

$$|s| \leq L, \quad \forall s \in \partial \phi_i(x), \forall x.$$

$\phi_i$ is said to be $(1/\gamma)$-smooth if it is differentiable and its derivative is $(1/\gamma)$-Lipschitz continuous.

We use $R \triangleq \max_{i \in [n]} \|a_i\|$ to denote the maximum row norm of $A$, and $R' \triangleq \max_{j \in [d]} \|a_j'\|$ to denote the maximum column norm of $A$, where $a_1', \ldots, a_d'$ are the columns of matrix $A$.

## 2 Stochastic Primal-Dual Method with $\mathcal{O}(1)$ Per-Iteration Cost

Our algorithm solves the following equivalent primal-dual reformulation of (1):

$$\min_{x \in X} \max_{y \in Y} \left\{ F(x, y) \triangleq \frac{1}{n} y^\top A x - \frac{1}{n} \sum_{i=1}^{n} \phi_i^*(y_i) + g(x) \right\}, \tag{5}$$

where $Y = Y_1 \times \cdots \times Y_n$ and $Y_i = \{y_i \in \mathbb{R}^n | \phi_i^*(y_i) < \infty\}$ is the dual feasible set resulted by the conjugate function $\phi_i^*$. For example, when $\phi_i$ is $L$-Lipschitz continuous, we have $Y_i \subset [-L, L]$ (see Lemma 4 in the Supplementary Materials.)

Our SPD1 algorithm for solving (5) is presented in Algorithm 1. In each iteration of the algorithm, only one coordinate of $x$ and one coordinate of $y$ are updated by using a randomly sampled data entry $a_{i_t j_t}$. Therefore, SPD1 only requires $\mathcal{O}(1)$ time per iteration. Because of this, SPD1 is a kind of randomized coordinate descent method.

---

**Algorithm 1** Stochastic Primal-Dual Method with $\mathcal{O}(1)$ Per-Iteration Cost (SPD1)

---

**Parameters:** primal step sizes $\{\eta_t\}$, dual step sizes $\{\tau_t\}$
Initialize $x^0 = \arg\min_{x \in X} g(x)$ and $y_i^0 = \arg\min_{y_i \in Y_i} \phi_i^*(y_i)$ for all $i \in [n]$
**for** $t = 0, 1, \cdots, T - 1$ **do**
    Randomly sample $i_t \in [n]$ and $j_t \in [d]$ independently and uniformly
$$x_j^{t+1} = \begin{cases} \text{prox}_{\eta_t g_j} \left( x_j^t - \eta_t \cdot a_{i_t j} y_{i_t}^t \right) & \text{if } j = j_t \\ x_j^t & \text{if } j \neq j_t \end{cases}$$
$$y_i^{t+1} = \begin{cases} \text{prox}_{(\tau_t/d)\phi_i^*} \left( y_i^t + \tau_t \cdot a_{i j_t} x_{j_t}^t \right) & \text{if } i = i_t \\ y_i^t & \text{if } i \neq i_t \end{cases}$$
**end for**
**Output:** $\hat{x}^T = \frac{1}{T} \sum_{t=0}^{T-1} x^t$ and $\hat{y}^T = \frac{1}{T} \sum_{t=0}^{T-1} y^t$

---

The intuition of this algorithm is as follows. One can rewrite (5) into:

$$\min_{x \in X} \max_{y \in Y} \left\{ F(x, y) = \frac{1}{n} \sum_{i=1}^{n} \sum_{j=1}^{d} \left[ a_{ij} y_i x_j - \frac{1}{d} \phi_i^*(y_i) + g_j(x_j) \right] \right\},$$

which has a two-layer finite-sum structure. Then Algorithm 1 can be viewed as a primal-dual version of stochastic gradient descent (SGD) on this finite-sum problem, which samples a pair of induces $(i_t, j_t)$ and then only utilizes the corresponding summand to do updates in each iteration. Hence, one can view SPD1 as a combination of randomized coordinate descent method and stochastic gradient descent method applied to the primal-dual reformulation (5).

Note that in the initialization stage, we need to minimize $g(x)$ and $\phi_i^*(y_i)$. Since we assume the proximal mappings of $g(x)$ and $\phi_i^*$ are easy to solve, these two direct minimization problems should be even easier, and thus would not bring any trouble in implementation of this algorithm. For example, when $g(x) = \lambda \|x\|_1$, it is well known its proximal mapping is the soft thresholding operator. While its direct minimizer, namely $x^0 = \min_x g(x)$, is simply $x^0 = 0$.

As a final remark, we point out that the primal-dual reformulation (5) is a convex-concave bilinear saddle point problem (SSP). This problem has drawn a lot of research attentions recently. For example, Chambolle and Pock developed an efficient primal-dual gradient method for solving bilinear SSP in [2], which has an accelerated rate in certain circumstances. Besides, in [3], Dang and Lan proposed a randomized algorithm for solving (5). However, their algorithm needs to use the full-dimensional gradient in each iteration, so the per-iteration cost is much higher than SPD1.

## 3 SPD1 with Variance Reduction

As we have discussed above, SPD1 has some close connection to SGD. Hence, we can incorporate the variance reduction technique [5] to reduce the variance of the stochastic gradients so that to improve the convergence rate of SPD1. This new algorithm, named SPD1-VR, is presented in Algorithm 2. Similar to SVRG [5], SPD1-VR has a two-loop structure. In the outer loop, the snapshots of the full

gradients are computed for both $\tilde{x}^k$ and $\tilde{y}^k$. In the inner loop, the updates are similar to Algorithm 1, but the stochastic gradient is replaced by its variance reduced version. That is, $a_{i_t j_t} y_{i_t}^t$ is replaced by $a_{i_t j_t}(y_{i_t}^t - \tilde{y}_{i_t}^k) + G_{x,j_t}^k$, where $G_{x,j_t}^k$ is the $j_t$-th coordinate of the latest snapshot of full gradient. This variance reduced stochastic gradient is still an unbiased estimator of the full gradient along direction $x_{j_t}$, i.e.,

$$\mathbb{E}_{i_t}\left[a_{i_t j_t}(y_{i_t}^t - \tilde{y}_{i_t}^k) + G_{x,j_t}^k\right] = \mathbb{E}_{i_t}\left[a_{i_t j_t} y_{i_t}^t\right] = \frac{1}{n}\sum_{i=1}^{n} a_{i j_t} y_i^t.$$

Because of the variance reduction technique, fixed step sizes $\eta$ and $\tau$ can be used instead of diminishing ones.

---

**Algorithm 2** SPD1 with Variance Reduction (SPD1-VR)

---

**Parameters**: primal step size $\eta$, dual step size $\tau$
Initialize $\tilde{x}^0 \in X$ and $\tilde{y}^0 \in Y$
**for** $k = 0, 1, \ldots, K-1$ **do**
    Compute full gradients $G_x^k = (1/n)A^\top \tilde{y}^k$ and $G_y^k = (1/d)A\tilde{x}^k$
    Let $(x^0, y^0) = (\tilde{x}^k, \tilde{y}^k)$
    **for** $t = 0, 1, \ldots, T-1$ **do**
        Randomly sample $i_t, i_t' \in [n]$ and $j_t, j_t' \in [d]$ independently
$$\bar{x}_j^t = \begin{cases} \text{prox}_{\eta g_j}\left[x_j^t - \eta \cdot \left(a_{i_t' j}(y_{i_t'}^t - \tilde{y}_{i_t'}^k) + G_{x,j}^k\right)\right] & \text{if } j = j_t \\ x_j^t & \text{if } j \neq j_t \end{cases}$$
$$\bar{y}_i^t = \begin{cases} \text{prox}_{(\tau/d)\phi_i^*}\left[y_i^t + \tau \cdot \left(a_{i j_t'}(x_{j_t'}^t - \tilde{x}_{j_t'}^k)\right) + G_{y,i}^k\right] & \text{if } i = i_t \\ y_i^t & \text{if } i \neq i_t \end{cases}$$
$$x_j^{t+1} = \begin{cases} \text{prox}_{\eta g_j}\left[x_j^t - \eta \cdot \left(a_{i_t j}(\bar{y}_{i_t}^t - \tilde{y}_{i_t}^k) + G_{x,j}^k\right)\right] & \text{if } j = j_t \\ x_j^t & \text{if } j \neq j_t \end{cases}$$
$$y_i^{t+1} = \begin{cases} \text{prox}_{(\tau/d)\phi_i^*}\left[y_i^t + \tau \cdot \left(a_{i j_t}(\bar{x}_{j_t}^t - \tilde{x}_{j_t}^k) + G_{y,i}^k\right)\right] & \text{if } i = i_t \\ y_i^t & \text{if } i \neq i_t \end{cases}$$
    **end for**
    Set $(\tilde{x}^{k+1}, \tilde{y}^{k+1}) = (x^T, y^T)$
**end for**
**Output:** $\tilde{x}^K$ and $\tilde{y}^K$

---

Besides the variance reduction technique, another crucial difference between SPD1 and SPD1-VR is that the latter is in fact an extragradient method [7]. Note that each iteration of the inner loop of SPD1-VR consists two gradient steps: the first step is a normal gradient descent/ascent, while in the second step, it starts from $x^t$ and $y^t$ but uses the gradient estimations at $(\bar{x}^t, \bar{y}^t)$. For saddle point problems, extragradient method has stronger convergence guarantees than simple gradient methods [8]. Moreover, in each iteration of SPD1-VR, two independent pairs of random indices $(i_t, j_t)$ and $(i_t', j_t')$ are drawn. This is because two stochastic gradients are needed for the extragradient framework. Similar to the classical analysis of stochastic algorithms, we need the stochastic gradients to be independent. However, when updating $\bar{x}^t$ and $x^{t+1}$, we choose the same coordinate $j_t$, so the independence property is only required for two directional stochastic gradients along coordinate $j_t$.

We note that every iteration of the inner loop only involves $\mathcal{O}(1)$ operations in SPD1-VR. Full gradients are computed in each outer loop, whose computational cost is $\mathcal{O}(nd)$.

Finally, we have to mention that [12] also developed a variance-reduction method for solving convex-concave saddle point problems, which is related to Algorithm 2. However, except for the common variance reduction ideas used in both methods, the method in [12] and SPD1-VR are quite different. First, there is no coordinate descent counterpart in their method, so the per-iteration cost is much higher than our method. Second, their method is a gradient method instead of extragradient method like SPD1-VR. Third, their method has quadratic dependence on the problem condition number unless extra acceleration technique is combined, while our method depends only linearly on condition number as shown in Section 4,.

## 4 Iteration Complexity Analysis

### 4.1 Iteration Complexity of SPD1

In this subsection, we analyze the convergence rate of SPD1 (Algorithm 1). We measure the optimality of the solution by primal-dual gap, which is defined as

$$\mathcal{G}(\hat{x}^T, \hat{y}^T) \triangleq \sup_{y \in Y} F(\hat{x}^T, y) - \inf_{x \in X} F(x, \hat{y}^T).$$

Note that primal-dual gap equals 0 if and only if $(\hat{x}^T, \hat{y}^T)$ is a pair of primal-dual optimal solutions to problem (5). Besides, primal-dual gap is always an upper bound of primal sub-optimality:

$$\mathcal{G}(\hat{x}^T, \hat{y}^T) \geq \sup_{y \in Y} F(\hat{x}^T, y) - \sup_{y \in Y} \inf_{x \in X} F(x, y)$$

$$\geq \sup_{y \in Y} F(\hat{x}^T, y) - \inf_{x \in X} \sup_{y \in Y} F(x, y)$$

$$= P(\hat{x}^T) - \inf_{x \in X} P(x).$$

Our main result for the iteration complexity of SPD1 is summarized in Theorem 1.

**Theorem 1.** *Assume each $\phi_i$ is $L$-Lipschitz continuous, and the primal feasible set $X$ is bounded, i.e.,*

$$D \triangleq \sup_{x \in X} \|x\| < \infty.$$

*If we choose the step sizes in SPD1 as*

$$\eta_t = \frac{\sqrt{2d}D}{LR\sqrt{t+1}} \quad and \quad \tau_t = \frac{\sqrt{2d}Ln}{DR'\sqrt{t+1}},$$

*then we have the following convergence rate for SPD1:*

$$\mathbb{E}\left[\mathcal{G}(\hat{x}^T, \hat{y}^T)\right] \leq \frac{\sqrt{2d}LD \cdot (R + R')}{\sqrt{T}}. \tag{6}$$

Note that when the problem is high-dimensional, i.e., $d \geq n$, it usually holds that $R \geq R'$. In this case, Theorem 1 implies that SPD1 requires

$$\mathcal{O}\left(\frac{dL^2 D^2 R^2}{\epsilon^2}\right) \tag{7}$$

iterations to ensure that the primal-dual gap is smaller than $\epsilon$.

Under the same assumptions, if we directly apply the classical result by Nemirovski et al. [9] for PSGD on the primal problem (1), the number of iterations needed by PSGD is

$$\mathcal{O}\left(\frac{L^2 D^2 R^2}{\epsilon^2}\right),$$

in order to reduce the primal sub-optimality to be smaller than $\epsilon$. Considering that each iteration of PSGD costs $\mathcal{O}(d)$ computation, its overall complexity is actually the same as (7).

If we further impose strong convexity and smoothness assumptions, we get an improved iteration complexity shown in Theorem 2.

**Theorem 2.** *We assume the same assumptions as in Theorem 1. Moreover, we assume that $g(x)$ is $\mu$-strongly convex ($\mu > 0$), and all $\phi_i$ are $(1/\gamma)$-smooth ($\gamma > 0$). If the step sizes in SPD1 are chosen as*

$$\eta_t = \frac{2}{\mu(t+4)} \quad and \quad \tau_t = \frac{2nd}{\gamma(t+4)},$$

*we have the following convergence rate for SPD1:*

$$\mathbb{E}\left[\mathcal{G}(\hat{x}^T, \hat{y}^T)\right] \leq \frac{4dD^2\mu + 4L^2\gamma + 2L^2R^2/\mu \cdot \ln(T+4) + 2dD^2R'^2/\gamma \cdot \ln(T+4)}{T}. \tag{8}$$

Comparing with the classical convergence rate result of PSGD, we note that the convergence rate of SPD1 not only depends on $\mu$, but also depends on the dual strong convexity parameter $\gamma$. This is reasonable because SPD1 has stochastic updates for both primal and dual variables, and this is why $\gamma > 0$ is necessary for ensuring the $\mathcal{O}(\ln T/T)$ convergence rate. Furthermore, we believe that the factor $\ln T$ is removable in (8) so $\mathcal{O}(1/T)$ convergence rate can be obtained, by applying more sophisticated analysis technique such as optimal averaging [15]. We do not dig into this to keep the paper more succinct.

## 4.2 Iteration Complexity of SPD1-VR

In this subsection, we analyze the iteration complexity of SPD1-VR (Algorithm 2). Before stating the main result, we first introduce the notion of condition number. When each $\phi_i$ is $(1/\gamma)$-smooth, and $g(x)$ is $\mu$-strongly convex, the condition number of the primal problem (1) defined in the literature is (see, e.g., [18]):

$$\kappa = \frac{R^2}{\mu\gamma}.$$

Here we also define another condition number:

$$\kappa' = \frac{dR'^2}{n\mu\gamma}.$$

Since $R$ is the maximum row norm and $R'$ is the maximum column norm of data matrix $A \in \mathbb{R}^{n \times d}$, usually $R^2$ and $(d/n)R'^2$ should be in the same order, which means $\kappa' \approx \kappa$. Without loss of generality, we assume that $\kappa \geq 1$ and $\kappa' \geq 1$.

**Theorem 3.** *Assume each $\phi_i$ is $(1/\gamma)$-smooth ($\gamma > 0$), and $g(x)$ is $\mu$-strongly convex ($\mu > 0$). If we choose the step sizes in SPD1-VR as*

$$\eta = \frac{\gamma}{128R^2} \cdot \min\left\{\frac{d\kappa}{n\kappa'}, 1\right\} \quad and \quad \tau = \frac{n\mu}{128R'^2} \cdot \min\left\{\frac{n\kappa'}{d\kappa}, 1\right\}, \tag{9}$$

*and let $T \geq c \cdot \max\{d\kappa, n\kappa'\}$ for some uniform constant $c > 0$ independent of problem setting, SPD1-VR converges linearly in expectation:*

$$\mathbb{E}\left[\Delta_K\right] \leq \left(\frac{3}{5}\right)^K \cdot \Delta_0,$$

*where*

$$\Delta_k = \frac{\|\tilde{x}^k - x^*\|^2}{\eta} + \frac{\|\tilde{y}^k - y^*\|^2}{\tau}$$

*and $(x^*, y^*)$ is a pair of primal-dual optimal solution to (5).*

This theorem implies that we need $\mathcal{O}(\log(1/\epsilon))$ outer loops, or $\mathcal{O}(\max\{d\kappa, n\kappa'\}\log(1/\epsilon))$ inner loops to ensure $\mathbb{E}[\Delta_k] \leq \epsilon$. Considering that there are $\Theta(nd)$ extra computation cost for computing the full gradient in every outer loop, the total complexity of this algorithm is

$$\mathcal{O}\left((nd + \max\{d\kappa, n\kappa'\})\log\frac{1}{\epsilon}\right). \tag{10}$$

As a comparison, the complexity of SVRG under same setting is [16]:

$$\mathcal{O}\left(d(n + \kappa)\log\frac{1}{\epsilon}\right), \tag{11}$$

Since usually it holds $\kappa' \approx \kappa$, $d\kappa$ will dominate $n\kappa'$ when $d > n$. In this case, the two complexity bounds (10) and (11) are the same.

Although the theoretical complexity of SPD1-VR is same as SVRG when $d \geq n$, we empirically found that SPD1-VR is significantly faster than SVRG in high-dimensional problems (see Section 5), by allowing much larger step sizes than the ones in (9) suggested by theory. We conjecture that this is due to the power of coordinate descent. Nesterov's seminal work [11] has rigorously proved that coordinate descent can reduce the Lipschitz constant of the problem, and thus allows larger step

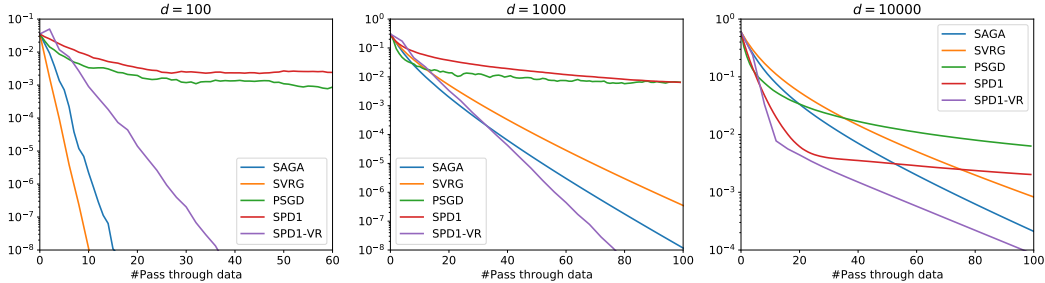

Figure 1: Experimental results on synthetic data. $n$ is set to 1000 in all figures, while $d$ varies from 100 to 10000. $\lambda$ is fixed as $\lambda = 10^{-3}$. The $y$-axis here is the primal sub-optimality, namely $P(x^t) - P(x^*)$.

sizes than gradient descent. However, due to the sophisticated coupling of primal and dual variable updates in our algorithm, our analysis is currently unable to reflect this property.

We point out that the existing accelerated algorithms such as SPDC [18] and Katyusha [1] have better complexity given by

$$\mathcal{O}\left(d(n + \sqrt{n\kappa}) \log \frac{1}{\epsilon}\right). \tag{12}$$

These accelerated algorithms employ Nesterov's extrapolation techniques [10] to accelerate the algorithms. We believe that it is also possible to incorporate the same technique to further accelerate SPD1-VR, but we leave this as a future work at this moment.

## 5  Experiments

In this section, we conduct numerical experiments of our proposed algorithms. Due to space limitation, we only present part of the results here, more experiments can be found in supplementary materials.

Here we consider solving a classification problem with logistic loss function

$$\phi_i(u) = \log\left(1 + \exp\{-b_i u\}\right),$$

where $b_i \in \{\pm 1\}$ is the class label. Note that this loss function is smooth. Although this $\phi_i^*$ does not admit closed-form solution for its proximal mapping, following [13], we apply Newton's method to compute its proximal mapping, which can achieve very high accuracy in very few (say, 5) steps. Since the proximal sub-problem of $\phi_i^*$ is a 1-dimensional optimization problem, using Newton's method here is actually quite cheap. Besides, We use $g(x) = \frac{\lambda}{2}\|x\|^2$ as the regularizer.

We compare our SPD1 and SPD1-VR with some standard stochastic algorithms for solving (1), including PSGD, SVRG and SAGA. We always set $T = nd$ for SPD1-VR and $T = n$ for SVRG, where $T$ is the number of inner loops in each outer loop.

### 5.1  Results on Synthetic Data

Since our theory in Section 4 suggest that the performance of our proposed methods relies on the relationship between $n$ and $d$. Here we will test our methods on synthetic dataset with different $n$ and $d$ to see their effects to the performance. To generate the data, we first randomly sample the data matrix $A$ and a vector $\bar{x} \in \mathbb{R}^d$ with entries i.i.d. drawn from $\mathcal{N}(0,1)$, and then generate the labels as

$$b_i = \text{sign}\left(a_i^\top \bar{x} + \varepsilon_i\right), \quad \varepsilon_i \sim \mathcal{N}(0, \sigma^2),$$

for some constant $\sigma^2 > 0$. Since the focus here is the relationship between $n$ and $d$, in order to simplify the experiments, we fix $n$ as $n = 1000$, but vary the value of $d$ with values chosen from $d \in \{100, 1000, 10000\}$.

The results are presented in Figure 1. When $d = 100 < n$, it is clear that SPD1 is slower than PSGD, and SPD1-VR is also inferior than both SVRG and SAGA. While when $d = 1000 = n$, even though SPD1 falls behind PSGD at the beginning, their final performance is quite close at last, and SPD1-VR

Table 1: Summary of datasets

| Dataset | $n$ | $d$ | $\lambda$ |
|---|---|---|---|
| colon-cancer | 62 | 2,000 | 1 |
| gisette | 6,000 | 5,000 | $10^{-2}$ |
| rcv1.binary | 20,242 | 47,236 | $10^{-3}$ |

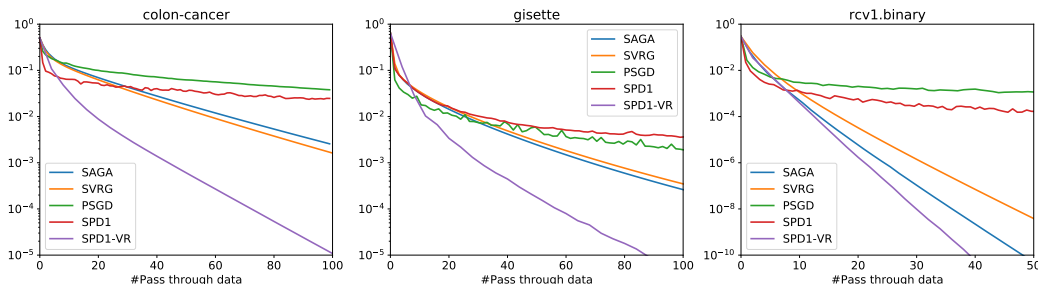

Figure 2: Numerical results on three real datasets. The $y$-axis is also primal sub-optimality.

begins to beat both SVRG and SAGA. Finally, when $d > n$, SPD1 becomes obviously faster than PSGD, and SPD1-VR is also significantly better than SVRG and SAGA. This indicates that our algorithms SPD1 and SPD1-VR are preferable in practice for high-dimensional problems.

## 5.2 Results on Real Datasets

In this part, we will demonstrate the efficiency of our proposed methods on real datasets. Here we only focus on the high-dimensional case where $d > n$ or $d \approx n$.

We will test all the algorithms on three real datasets: `colon-cancer`, `gisette` and `rcv1.binary`, downloaded from the LIBSVM website [2]. The attributes of these data and $\lambda$ used for each dataset are summarized in Table 1.

The experimental results on these real datasets are shown in Figure 2. For `colon-cancer` dataset, where $d$ is much larger than $n$, the performance of SPD1-VR is really dominating over other methods, and SPD1 also performs better than PSGD. For `gisette` dataset, where $n$ is slightly larger than $d$, SPD1-VR still outperforms all other competitors, but this time SPD1 is slower than PSGD. Besides, for `rcv1.binary`, both SPD1 and SPD1-VR are better than PSGD and SVRG/SAGA respectively.

These results on real datasets further confirm that our proposed methods, especially SPD1-VR, are faster than existing algorithms on high-dimensional problems.

## 6 Conclusion

In this paper, we developed two stochastic primal-dual algorithms, named SPD1 and SPD1-VR for solving regularized empirical risk minimization problems. Different from existing methods, our proposed algorithms have a brand-new updating style, which only need to use one coordinate of one sampled data in each iteration. As a result, the per-iteration cost is very low and the algorithms are very suitable for distributed systems. We proved that the overall convergence property of SPD1 and SPD1-VR resembles PSGD and SVRG respectively under certain condition, and empirically showed that they are faster than existing methods such as PSGD, SVRG and SAGA in high-dimensional settings. We believe that our new methods have great potential to be used in large-scale distributed optimization applications.

# 7 Acknowledgement

S. Ma is partly supported by a startup package in Department of Mathematics at UC Davis and the National Natural Science Foundation of China under Grant 11631013. J. Liu is in part supported by NSF CCF1718513, IBM faculty award, and NEC fellowship.

## Footnotes

[2]`www.csie.ntu.edu.tw/~cjlin/libsvmtools/datasets`

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
