[Supplementary Material]

# Supplementary Materials: Stochastic Primal-Dual Method for Empirical Risk Minimization with $\mathcal{O}(1)$ Per-Iteration Complexity

**Conghui Tan**[*]
The Chinese University of Hong Kong
chtan@se.cuhk.edu.hk

**Tong Zhang**
Tencent AI Lab
tongzhang@tongzhang-ml.org

**Shiqian Ma**
University of California, Davis
sqma@math.ucdavis.edu

**Ji Liu**
Tencent AI Lab, University of Rochester
ji.liu.uwisc@gmail.com

## 1 Proofs

**Lemma 4.** *If $\phi : \mathbb{R} \to \mathbb{R}$ is L-Lipschitz continuous, then $\{y \in \mathbb{R} | \phi^*(y) < \infty\} \subseteq [-L, L]$.*

*Proof.* In this proof, we will first show that if $y > L$, then $\phi^*(y) = \infty$. By the Lipschitz continuity, we have

$$\phi(x) \le \phi(0) + L|x|.$$

Plug it into the definition of convex conjugate (4), we can obtain

$$\phi^*(y) = \sup_{x \in \mathbb{R}} \{y \cdot x - \phi(x)\} \ge \sup_{x \in \mathbb{R}} \{y \cdot x - L|x|\} - \phi(0).$$

Since $y > L$, we can always let $(y \cdot x - L|x|)$ goes to infinity by set $x \to +\infty$, which means $\phi^*(y) = \infty$. And the proof for the case $y < -L$ is similar. $\square$

### 1.1 Proofs Concerning Algorithm 1

In this part, we will use

$$\mathcal{F}_t = \{i_0, j_0, , \ldots, i_{t-1}, j_{t-1}\}$$

to denote all the random variables generated before iteration $t$. And for simplicity, we will denote

$$\phi^*(y) = \sum_{i=1}^{n} \phi_i^*(y_i).$$

The following lemma is the key to prove both Theorem 1 and Theorem 2.

**Lemma 5.** *Consider t-th iteration of Algorithm 1. Assume $g(x)$ is $\mu$-strongly convex ($\mu \ge 0$, and $\mu = 0$ means general convexity), and all $\phi_i$ are $(1/\gamma)$-smooth ($\gamma \ge 0$). When conditioned on $\mathcal{F}_t$, it*

---

[*]This work was done while Conghui Tan was a research intern at Tencent AI lab.

*holds that*

$$\frac{d}{2\eta_t} \cdot \left[\left(1 - \min\left\{\frac{\eta_t\mu}{2}, \frac{1}{4}\right\}\right) \|x^t - x\|^2 - \mathbb{E}\left[\|x^{t+1} - x\|^2 | \mathcal{F}_t\right]\right] +$$

$$\frac{d}{2\tau_t} \cdot \left[\left(1 - \min\left\{\frac{\tau_t\gamma}{2d}, \frac{1}{4}\right\}\right) \|y^t - y\|^2 - \mathbb{E}\left[\|y^{t+1} - y\|^2 | \mathcal{F}_t\right]\right]$$

$$\geq F(x^t, y) - F(x, y^t) - \eta_t L^2 R^2 - \frac{\eta_t D^2 R'^2}{n}$$

$$+ d \cdot \left[\mathbb{E}\left[g(x^{t+1}) | \mathcal{F}_t\right] - g(x^t)\right] + \mathbb{E}\left[\phi^*(y^{t+1}) | \mathcal{F}_t\right] - \phi^*(y^t) \tag{13}$$

*for any $x \in X$ and $y \in Y$.*

*Proof.* Here we define two "imaginary" iterates $x'^t$ and $y'^t$ in the following way:

$$x_j'^t = \mathrm{prox}_{\eta_t g_j}\left(x_j^t - \eta_t a_{i_t j} y_{i_t}^t\right) \quad \forall j \in [d],$$
$$y_i'^t = \mathrm{prox}_{(\tau_t/d)\phi_i^*}\left(y_i^t - \tau_t a_{ij_t} x_{j_t}^t\right) \quad \forall i \in [n].$$

Note that $x_j^{t+1} = x_j'^t$ when $j = j_t$, and $x_j^{t+1} = x_j^t$ otherwise. And $y_i^{t+1} = y_i'^t$ when $i = i_t$, while $y_i^{t+1} = y_i^t$ when $i \neq i_t$.

For each $j \in [d]$, due to $(a+b)^2 - a^2 - b^2 = 2a \cdot b$, we have

$$(x_j^t - x_j)^2 - (x_j'^t - x_j)^2 - (x_j^t - x_j'^t)^2 = 2(x_j^t - x_j'^t) \cdot (x_j'^t - x_j)$$

for any $x_j \in X_j$. By the definition of $x_j'^t$ and the optimality condition of the proximal subproblem, there must exist a subgradient $s \in \partial g_j(x_j'^t) + \partial \mathbb{1}_{X_j}(x_j'^t)$ such that

$$x_j'^t = x_j^t - \eta_t a_{i_t j} y_{i_t}^t - \eta_t s,$$

where $\mathbb{1}_{X_j}$ is the indicator function of set $X_j$, i.e.,

$$\mathbb{1}_X(x) = \begin{cases} 0, & \text{if } x \in X, \\ +\infty, & \text{if } x \notin X. \end{cases}$$

Plug in this fact into the previous equality, we have

$$(x_j^t - x_j)^2 - (x_j'^t - x_j)^2 - (x_j^t - x_j'^t)^2 = 2\eta_t(a_{i_t j} y_{i_t}^t + s) \cdot (x_j'^t - x_j).$$

Since $g(x)$ is $\mu$-strongly convex, and also because of the separable assumption (2), we know that each $g_j(x_j)$ is also $\mu$-strongly convex. Then we apply strong convexity to subgradient $s$:

$$(x_j^t - x_j)^2 - (x_j'^t - x_j)^2 - (x_j^t - x_j'^t)^2$$
$$= 2\eta(a_{i_t j} y_{i_t}^t + s) \cdot (x_j'^t - x_j)$$
$$\geq 2\eta_t a_{i_t j} y_{i_t}^t \cdot (x_j'^t - x_j) + 2\eta_t \left[g_j(x_j'^t) - g_j(x_j) + \frac{\mu}{2}(x_j'^t - x_j)^2 + \mathbb{1}_{X_j}(x_j'^t) - \mathbb{1}_{X_j}(x_j)\right].$$

Since $x'^t$ and $x$ are always feasible by definition, thus $\mathbb{1}_{X_j}(x_j'^t) = \mathbb{1}_{X_j}(x_j) = 0$. As a result,

$$(x_j^t - x_j)^2 - (x_j'^t - x_j)^2 - (x_j^t - x_j'^t)^2$$
$$\geq 2\eta_t a_{i_t j} y_{i_t}^t \cdot (x_j'^t - x_j) + 2\eta_t \left[g_j(x_j'^t) - g_j(x_j) + \frac{\mu}{2}(x_j'^t - x_j)^2\right]$$
$$= 2\eta_t a_{i_t j} y_{i_t}^t \cdot (x_j^t - x_j) + 2\eta_t a_{i_t j} y_{i_t}^t \cdot (x_j'^t - x_j^t) + 2\eta_t[g_j(x_j'^t) - g_j(x_j)] + \eta_t \mu(x_j'^t - x_j)^2$$
$$\geq 2\eta_t a_{i_t j} y_{i_t}^t \cdot (x_j^t - x_j) - 2\eta_t^2 \left(a_{i_t j} y_{i_t}^t\right)^2 - \frac{1}{2}(x_j^t - x_j'^t)^2 + 2\eta_t[g_j(x_j'^t) - g_j(x_j)] + \eta_t \mu(x_j'^t - x_j)^2,$$

where the last line is due to Cauchy-Schwarz inequality. By cancelling term $(1/2)(x_j^t - x_j'^t)^2$ on both sides, and note that

$$\frac{1}{2}(x_j^t - x_j'^t)^2 + \eta_t \mu(x_j'^t - x_j)^2 \geq \min\left\{\frac{\eta_t\mu}{2}, \frac{1}{4}\right\}(x_j^t - x_j)^2,$$

we can further rewrite the above inequality as

$$\left(1 - \min\left\{\frac{\eta_t\mu}{2}, \frac{1}{4}\right\}\right)(x_j^t - x_j)^2 - (x_j'^t - x_j)^2$$

$$\geq 2\eta_t a_{i_t j} y_{i_t}^t \cdot (x_j^t - x_j) - 2\eta_t^2 \left(a_{i_t j} y_{i_t}^t\right)^2 + 2\eta_t[g_j(x_j'^t) - g_j(x_j)].$$

We can bound the second term on the right-hand-side by $(a_{i_t j} y_{i_t})^2 \leq L^2 a_{i_t j}^2$ because of Lemma 4, and we divide both sides by $2\eta_t$, then we obtain

$$\frac{1}{2\eta_t}\left[\left(1 - \min\left\{\frac{\eta_t\mu}{2}, \frac{1}{4}\right\}\right)(x_j^t - x_j)^2 - (x_j'^t - x_j)^2\right] \geq a_{i_t j} y_{i_t}^t \cdot (x_j^t - x_j) + g_j(x_j'^t) - g_j(x_j) - \eta_t L^2 a_{i_t j}^2. \tag{14}$$

Even though $x_j'^t$ depends on $i_t$, it is independent of $j_t$. We observe that $x_j^{t+1}$ takes value $x_j'^t$ with probability $1/d$, and $x_j^t$ otherwise. Hence, by conditioning on $\mathcal{F}_t' \triangleq \mathcal{F}_t \cup \{i_t\}$, we always have

$$\mathbb{E}\left[g_j(x_j^{t+1})|\mathcal{F}_t'\right] = \frac{1}{d}g_j(x_j'^t) + \frac{d-1}{d}g_j(x_j^t),$$

$$\mathbb{E}\left[\|x_j^{t+1} - x_j\|^2|\mathcal{F}_t'\right] = \frac{1}{d}\|x_j'^t - x_j\|^2 + \frac{d-1}{d}\|x_j^t - x_j\|^2.$$

Put these relationships into (14), when conditioned on $\mathcal{F}_t'$, we obtain:

$$\frac{d}{2\eta_t} \cdot \left[\left(1 - \min\left\{\frac{\eta_t\mu}{2}, \frac{1}{4}\right\}\right)(x_j^t - x_j)^2 - \mathbb{E}\left[(x_j^{t+1} - x_j)^2|\mathcal{F}_t'\right]\right]$$

$$\geq a_{i_t j} y_{i_t}^t \cdot (x_j^t - x_j) + d \cdot \mathbb{E}\left[g_j(x_j^{t+1})|\mathcal{F}_t'\right] - (d-1)g_j(x_j^t) - g_j(x_j) - \eta_t L^2 a_{i_t j}^2.$$

We sum this inequality from $j = 1$ to $j = d$, and finally obtain:

$$\frac{d}{2\eta_t} \cdot \left[\left(1 - \min\left\{\frac{\eta_t\mu}{2}, \frac{1}{4}\right\}\right)\|x^t - x\|^2 - \mathbb{E}\left[\|x^{t+1} - x\|^2|\mathcal{F}_t'\right]\right]$$

$$\geq \sum_{j=1}^d a_{i_t j} y_{i_t}^t \cdot (x_j^t - x_j) + d \cdot \mathbb{E}\left[g(x^{t+1})|\mathcal{F}_t'\right] - (d-1)g(x^t) - g(x) - \eta_t L^2 \sum_{j=1}^d a_{i_t j}^2$$

$$\geq \sum_{j=1}^d a_{i_t j} y_{i_t}^t \cdot (x_j^t - x_j) + d \cdot \mathbb{E}\left[g(x^{t+1})|\mathcal{F}_t'\right] - (d-1)g(x^t) - g(x) - \eta_t L^2 R^2,$$

where the last inequality is due to the definition $R = \max_i \|a_i\|$. Now, we further take expectation with respect to $i_t$:

$$\frac{d}{2\eta_t} \cdot \left[\left(1 - \min\left\{\frac{\eta_t\mu}{2}, \frac{1}{4}\right\}\right)\|x^t - x\|^2 - \mathbb{E}\left[\|x^{t+1} - x\|^2|\mathcal{F}_t\right]\right]$$

$$\geq \frac{1}{n}y^{t\top}A(x^t - x) + d \cdot \mathbb{E}\left[g(x^{t+1})|\mathcal{F}_t\right] - (d-1)g(x^t) - g(x) - \eta_t L^2 R^2. \tag{15}$$

On the other hand, due to the symmetric nature of Algorithm 1, and noticing $(1/\gamma)$-smoothness of $\phi_i$ implies $\gamma$-strong convexity of $\phi_i^*$, we can derive similar inequality for $y^{t+1}$:

$$\frac{n}{2\tau_t} \cdot \left[\left(1 - \min\left\{\frac{\tau_t\gamma}{2d}, \frac{1}{4}\right\}\right)\|y^t - y\|^2 - \mathbb{E}\left[\|y^{t+1} - y\|^2|\mathcal{F}_t\right]\right]$$

$$\geq \frac{1}{d}(y - y^t)Ax^t + \frac{n}{d} \cdot \mathbb{E}\left[\phi^*(y^{t+1})|\mathcal{F}_t\right] - \frac{n-1}{d}\phi^*(y^t) - \frac{1}{d}\phi^*(y) - \frac{\tau_t D^2 R'^2}{d},$$

or equivalently,

$$\frac{d}{2\tau_t} \cdot \left[\left(1 - \min\left\{\frac{\tau_t\gamma}{2d}, \frac{1}{4}\right\}\right)\|y^t - y\|^2 - \mathbb{E}\left[\|y^{t+1} - y\|^2|\mathcal{F}_t\right]\right]$$

$$\geq \frac{1}{n}(y - y^t)Ax^t + \mathbb{E}\left[\phi^*(y^{t+1})|\mathcal{F}_t\right] - \frac{n-1}{n}\phi^*(y^t) - \frac{1}{n}\phi^*(y) - \frac{\tau_t D^2 R'^2}{n},$$

Let us add this inequality with (15):

$$\frac{d}{2\eta_t} \cdot \left[ \left(1 - \frac{\eta_t \mu}{2}\right) \|x^t - x\|^2 - \mathbb{E}\left[\|x^{t+1} - x\|^2 | \mathcal{F}_t\right] \right] +$$

$$\frac{d}{2\tau_t} \cdot \left[ \left(1 - \min\left\{\frac{\tau_t \gamma}{2d}, \frac{1}{4}\right\}\right) \|y^t - y\|^2 - \mathbb{E}\left[\|y^{t+1} - y\|^2 | \mathcal{F}_t\right] \right]$$

$$= \frac{1}{n} y^{t\top} A(x^t - x) + \frac{1}{n}(y - y^t) A x^t + d \cdot \mathbb{E}\left[g(x^{t+1}) | \mathcal{F}_t\right] - (d-1)g(x^t) - g(x)$$

$$+ \mathbb{E}\left[\phi^*(y^{t+1}) | \mathcal{F}_t\right] - \frac{n-1}{n}\phi^*(y^t) - \frac{1}{n}\phi^*(y) - \eta_t L^2 R^2 - \frac{\eta_t D^2 R'^2}{n}$$

$$= \left[\frac{1}{n} y^\top A x^t + g(x^t) - \frac{1}{n}\phi^*(y)\right] - \left[\frac{1}{n} y^{t\top} A x + g(x) - \frac{1}{n}\phi^*(y^t)\right] - \eta_t L^2 R^2 - \frac{\eta_t D^2 R'^2}{n}$$

$$+ d \cdot \left[\mathbb{E}\left[g(x^{t+1}) | \mathcal{F}_t\right] - g(x^t)\right] + \mathbb{E}\left[\phi^*(y^{t+1}) | \mathcal{F}_t\right] - \phi^*(y^t)$$

$$= F(x^t, y) - F(x, y^t) - \eta_t L^2 R^2 - \frac{\eta_t D^2 R'^2}{n}$$

$$+ d \cdot \left[\mathbb{E}\left[g(x^{t+1}) | \mathcal{F}_t\right] - g(x^t)\right] + \mathbb{E}\left[\phi^*(y^{t+1}) | \mathcal{F}_t\right] - \phi^*(y^t),$$

which is our desired result. $\qquad\square$

**Lemma 6.** *Under same assumption as Lemma 5, it holds that*

$$\frac{d}{2} \cdot \sum_{t=0}^{T-1} \frac{\mathbb{E}\left[\left(1 - \min\left\{\frac{\eta_t \mu}{2}, \frac{1}{4}\right\}\right) \|x^t - x\|^2 - \|x^{t+1} - x\|^2\right]}{\eta_t} +$$

$$\frac{d}{2} \cdot \sum_{t=0}^{T-1} \frac{\mathbb{E}\left[\left(1 - \min\left\{\frac{\tau_t \gamma}{2d}, \frac{1}{4}\right\}\right) \|y^t - y\|^2 - \|y^{t+1} - y\|^2\right]}{\tau_t}$$

$$\geq T \cdot \mathbb{E}\left[F(\hat{x}^T, y) - F(x, \hat{y}^T)\right] - L^2 R^2 \sum_{t=0}^{T-1} \eta_t - \frac{D^2 R'^2}{n} \sum_{t=0}^{T-1} \tau_t. \tag{16}$$

*for Algorithm 1.*

*Proof.* By summing up inequality (13) from $t = 0$ to $t = T - 1$, and applying the law of total expectation, we have

$$\frac{d}{2} \cdot \sum_{t=0}^{T-1} \frac{\mathbb{E}\left[\left(1 - \min\left\{\frac{\eta_t \mu}{2}, \frac{1}{4}\right\}\right) \|x^t - x\|^2 - \|x^{t+1} - x\|^2\right]}{\eta_t}$$

$$+ \frac{d}{2} \cdot \sum_{t=0}^{T-1} \frac{\mathbb{E}\left[\left(1 - \min\left\{\frac{\tau_t \gamma}{2d}, \frac{1}{4}\right\}\right) \|y^t - y\|^2 - \|y^{t+1} - y\|^2\right]}{\tau_t}$$

$$\geq \sum_{t=0}^{T-1} \mathbb{E}\left[F(x^t, y) - F(x, y^t)\right] - L^2 R^2 \sum_{t=0}^{T-1} \eta_t - \frac{D^2 R'^2}{n} \sum_{t=0}^{T-1} \tau_t$$

$$+ d \sum_{t=0}^{T-1} \mathbb{E}\left[g(x^{t+1}) - g(x^t)\right] + \sum_{t=0}^{T-1} \mathbb{E}\left[\phi^*(y^{t+1}) - \phi^*(y^t)\right]$$

$$= \sum_{t=0}^{T-1} \mathbb{E}\left[F(x^t, y) - F(x, y^t)\right] - L^2 R^2 \sum_{t=0}^{T-1} \eta_t - \frac{D^2 R'^2}{n} \sum_{t=0}^{T-1} \tau_t$$

$$+ d \cdot \mathbb{E}\left[g(x^T) - g(x^0)\right] + \mathbb{E}\left[\phi^*(y^T) - \phi^*(y^0)\right].$$

Recall that $x^0$ and $y^0$ satisfy $x^0 = \arg\min_{x \in X} g(x)$ and $y^0 = \arg\min_{y \in Y} \phi^*(y)$, which means $g(x^T) - g(x^0) \geq 0$ and $\phi^*(y^T) - \phi^*(y^0) \geq 0$. Hence,

$$\frac{d}{2} \cdot \sum_{t=0}^{T-1} \frac{\mathbb{E}\left[\left(1 - \min\left\{\frac{\eta_t \mu}{2}, \frac{1}{4}\right\}\right) \|x^t - x\|^2 - \|x^{t+1} - x\|^2\right]}{\eta_t}$$

$$+ \frac{d}{2} \cdot \sum_{t=0}^{T-1} \frac{\mathbb{E}\left[\left(1 - \min\left\{\frac{\tau_t \gamma}{2d}, \frac{1}{4}\right\}\right) \|y^t - y\|^2 - \|y^{t+1} - y\|^2\right]}{\tau_t}$$

$$\geq \sum_{t=0}^{T-1} \mathbb{E}\left[F(x^t, y) - F(x, y^t)\right] - dL^2 R^2 \sum_{t=0}^{T-1} \eta_t - \frac{D^2 R'^2}{n} \sum_{t=0}^{T-1} \tau_t. \qquad (17)$$

Besides, we use the definition of $\bar{x}^T$ and $\bar{y}^T$, along with the convexity-concavity of $F(x, y)$, we can obtain

$$T \cdot \mathbb{E}\left[F(\hat{x}^T, y) - F(x, \hat{y}^T)\right] \leq \sum_{t=0}^{T-1} \mathbb{E}\left[F(x^t, y) - F(x, y^t)\right].$$

Now we can complete the proof by combining this fact with (17). $\qquad \square$

Now, we are ready to prove Theorem 1:

*Proof of Theorem 1.* Lemma 6 can be applied here with $\mu = 0$ and $\gamma = 0$. Let us first bound the term on the left-hand-side of (16):

$$\sum_{t=0}^{T-1} \frac{\mathbb{E}\left[\|x^t - x\|^2 - \|x^{t+1} - x\|^2\right]}{\eta_t}$$

$$= \frac{1}{\eta_0} \cdot \|x^0 - x\|^2 + \sum_{t=1}^{T-1} \left(\frac{1}{\eta_t} - \frac{1}{\eta_{t-1}}\right) \mathbb{E}\left[\|x^t - x\|^2\right] - \frac{1}{\eta_{T-1}} \cdot \mathbb{E}\left[\|x^T - x\|^2\right]$$

$$\leq \frac{1}{\eta_0} \cdot 4D^2 + \sum_{t=1}^{T-1} \left(\frac{1}{\eta_t} - \frac{1}{\eta_{t-1}}\right) 4D^2$$

$$= \frac{4D^2}{\eta_{T-1}}$$

by the boundness assumption of $X$, along with the fact sequence $\{\eta_t\}$ is non-increasing. By applying Lemma 4 again, we know that $\|y^t - y\|^2 \leq 2nL^2$, and hence we can similarly show that

$$\sum_{t=0}^{T-1} \frac{\mathbb{E}\left[\|y^t - y\|^2 - \|y^{t+1} - y\|^2\right]}{\tau_t} \leq \frac{4nL^2}{\tau_{T-1}}.$$

Combine all these facts into (16), one can obtain

$$T \cdot \mathbb{E}\left[F(\hat{x}^T, y) - F(x, \hat{y}^T)\right] \leq \frac{2dD^2}{\eta_{T-1}} + \frac{2ndL^2}{\tau_{T-1}} + L^2 R^2 \sum_{t=0}^{T-1} \eta_t + \frac{D^2 R'^2}{n} \sum_{t=0}^{T-1} \tau_t.$$

Divide both sides by $T$, and then take supremum with respect to $x \in X$ and $y \in Y$:

$$\mathbb{E}\left[\mathcal{G}(\hat{x}^T, \hat{y}^T)\right]$$

$$= \sup_{x \in X, y \in Y} \mathbb{E}\left[F(\hat{x}^T, y) - F(x, \hat{y}^T)\right]$$

$$\leq \frac{2dD^2/\eta_{T-1} + 2ndL^2/\tau_{T-1} + L^2 R^2 \sum_{t=0}^{T-1} \eta_t + (D^2 R'^2/n) \sum_{t=0}^{T-1} \tau_t}{T},$$

and the theorem can be proven by plug the values of $\{\eta_t\}$ and $\{\tau_t\}$ into this inequality. $\qquad \square$

We are ready to prove Theorem 2:

*Proof of Theorem 2.* Again, we need to apply Lemma 6 here. The first term on the left-hand-side of (16) can be bounded in the following way:

$$\sum_{t=0}^{T-1} \frac{\mathbb{E}\left[\left(1 - \min\left\{\frac{\eta_t \mu}{2}, \frac{1}{4}\right\}\right) \|x^t - x\|^2 - \|x^{t+1} - x\|^2\right]}{\eta_t}$$

$$= \frac{1 - \min\left\{\frac{\eta_0 \mu}{2}, \frac{1}{4}\right\}}{\eta_0} \cdot \|x^0 - x\|^2 + \sum_{t=1}^{T-1} \left(\frac{1 - \min\left\{\frac{\eta_t \mu}{2}, \frac{1}{4}\right\}}{\eta_t} - \frac{1}{\eta_{t-1}}\right) \mathbb{E}\left[\|x^t - x\|^2\right] - \frac{1}{\eta_{T-1}} \cdot \mathbb{E}\left[\|x^T - x\|^2\right]$$

$$\leq \frac{1}{\eta_0} \cdot \|x^0 - x\|^2 + \sum_{t=1}^{T-1} \max\left\{\frac{1}{\eta_t} - \frac{1}{\eta_{t-1}} - \frac{\mu}{2}, \frac{3}{4\eta_t} - \frac{1}{\eta_{t-1}}\right\} \mathbb{E}\left[\|x^t - x\|^2\right]$$

$$\leq \frac{1}{\eta_0} \cdot \|x^0 - x\|^2 + \sum_{t=1}^{T-1} 0 \cdot \mathbb{E}\left[\|x^t - x\|^2\right]$$

$$\leq \frac{4}{\eta_0} \cdot D^2,$$

where the third equality follows from our choices of $\{\eta_t\}$, and the last inequality is due to boudness assumption again. Similarly, we can derive the bound

$$\sum_{t=0}^{T-1} \frac{\mathbb{E}\left[\left(1 - \min\left\{\frac{\tau_t \gamma}{2d}, \frac{1}{4}\right\}\right) \|y^t - y\|^2 - \|y^{t+1} - y\|^2\right]}{\tau_t} \leq \frac{4nL^2}{\tau_0}.$$

Combine these two bounds into (16) and then take supremum with respect to $x$ and $y$, we can finally obtain:

$$\mathbb{E}\left[\mathcal{G}(\hat{x}^T, \hat{y}^T)\right] \leq \frac{2dD^2/\eta_0 + 2ndL^2/\tau_0 + L^2 R^2 \sum_{t=0}^{T-1} \eta_t + (D^2 R'^2/n) \sum_{t=0}^{T-1} \tau_t}{T}.$$

We can finish the proof by plug the values of $\{\eta_t\}$ and $\{\tau_t\}$. $\qquad\square$

## 1.2 Proofs Concerning Algorithm 2

Again, we use
$$\mathcal{F}_t = \{i_0, i_0', j_0, j_0', \ldots, i_{t-1}, i_{t-1}', j_{t-1}, j_{t-1}'\}$$
to denote all the random variables generated before iteration $t$. We use the following notation to denote the variance-reduced gradients used in our algorithms:

$$\nabla_{x,j}^t = a_{i_t'j}(y_{i_t'}^t - \tilde{y}_{i_t'}^k) + G_{x,j}^k,$$
$$\nabla_{y,i}^t = a_{ij_t'}(x_{j_t'}^t - \tilde{x}_{j_t'}^k) + G_{y,i}^k,$$
$$\nabla_{x,j}'^t = a_{i_t j}(\bar{y}_{i_t}^t - \tilde{y}_{i_t}^k) + G_{x,j}^k,$$
$$\nabla_{y,i}'^t = a_{i j_t}(\bar{x}_{j_t}^t - \tilde{x}_{j_t}^k) + G_{y,i}^k.$$

Besides, we will also define two "imaginary" iterates $x'^t$ and $y'^t$:

$$x_j'^t = \text{prox}_{\eta g_j} \left(x_j^t - \eta \nabla_{x,j}^t\right) \quad \forall j \in [d],$$
$$y_i'^t = \text{prox}_{(\tau/d)\phi_i^*} \left(y_i^t - \tau \nabla_{y,i}^t\right) \quad \forall i \in [n].$$

Obviously, $\bar{x}_j^t = x_j'^t$ when $j = j_t$ and $\bar{y}_i^t = y_i'^t$ when $i = i_t$. A key observation here is that each $x_j'^t$ only depends on $i_t'$ when conditioned on $\mathcal{F}_t$, and it is independent of $i_t$, $j_t$ and $j_t'$. Similarly, $y_i'^t$ is also independent of $i_t$, $j_t$ and $i_t'$.

First, let us develop a bound for the gradient variance:

**Lemma 7.** *Algorithm 2 has the following bounds for its gradient variance:*

$$\mathbb{E}\left[(\nabla_{x,j_t}^t - \nabla_{x,j_t}'^t)^2 | \mathcal{F}_t\right] \le \frac{8R^2}{nd}\|y - \tilde{y}^k\|^2 + \frac{12R^2}{nd}\|y^t - y\|^2 + \frac{8R^2}{d}\mathbb{E}\left[\|y^t - \bar{y}^t\|^2 | \mathcal{F}_t\right] \quad (18)$$

$$\mathbb{E}\left[(\nabla_{y,i_t}^t - \nabla_{y,i_t}'^t)^2 | \mathcal{F}_t\right] \le \frac{8R'^2}{nd}\|x - \tilde{x}^k\|^2 + \frac{12R'^2}{nd}\|x^t - x\|^2 + \frac{8R'^2}{n}\mathbb{E}\left[\|x^t - \bar{x}^t\|^2 | \mathcal{F}_t\right] \quad (19)$$

*for any $x \in X$ and $y \in Y$.*

*Proof.* Just by definition of $\nabla_{x,j_t}^t$ and $\nabla_{x,j_t}'^t$, and repeatedly using $(a+b)^2 \le 2a^2 + 2b^2$, we have

$$(\nabla_{x,j_t}^t - \nabla_{x,j_t}'^t)^2 = \left[a_{i_t'j_t}(y_{i_t'}^t - \tilde{y}_{i_t'}^k) - a_{i_t j_t}(\bar{y}_{i_t}^t - \tilde{y}_{i_t}^k)\right]^2$$
$$\le 2a_{i_t'j_t}^2(y_{i_t'}^t - \tilde{y}_{i_t'}^k)^2 + 2a_{i_t j_t}^2(\bar{y}_{i_t}^t - \tilde{y}_{i_t}^k)^2$$
$$\le 2a_{i_t'j_t}^2\left[2(y_{i_t'}^t - y_{i_t'})^2 + 2(y_{i_t'} - \tilde{y}_{i_t'}^k)^2\right]$$
$$\quad + 2a_{i_t j_t}^2\left[2(\tilde{y}_{i_t}^k - y_{i_t})^2 + 2(y_{i_t} - \bar{y}_{i_t}^t)^2\right]$$
$$\le 2a_{i_t'j_t}^2\left[2(y_{i_t'} - y_{i_t'}^t)^2 + 2(y_{i_t'} - \tilde{y}_{i_t'}^k)^2\right]$$
$$\quad + 2a_{i_t j_t}^2\left[2(\tilde{y}_{i_t}^k - y_{i_t})^2 + 4(y_{i_t} - y_{i_t}^t)^2 + 4(y_{i_t}^t - \bar{y}_{i_t}^t)^2\right]$$
$$= 2a_{i_t'j_t}^2\left[2(y_{i_t'} - y_{i_t'}^t)^2 + 2(y_{i_t'} - \tilde{y}_{i_t'}^k)^2\right]$$
$$\quad + 2a_{i_t j_t}^2\left[2(\tilde{y}_{i_t}^k - y_{i_t})^2 + 4(y_{i_t} - y_{i_t}^t)^2 + 4\|y^t - \bar{y}^t\|^2\right], \quad (20)$$

where the last line is due to the fact that $\bar{x}^t$ and $x^t$ only differ in coordinate $i_t$. In the next, we will take expectation. Since $i_t'$ and $j_t$ are independent, then

$$\mathbb{E}\left[a_{i_t'j_t}^2(y_{i_t'} - y_{i_t'}^t)^2 | \mathcal{F}_t\right] = \mathbb{E}\left[\mathbb{E}_{j_t}\left[a_{i_t'j_t}^2\right] \cdot (y_{i_t'} - y_{i_t'}^t)^2 | \mathcal{F}_t\right]$$
$$= \mathbb{E}\left[\frac{\|a_{i_t}\|^2}{d} \cdot (y_{i_t'} - y_{i_t'}^t)^2 | \mathcal{F}_t\right]$$
$$\le \frac{R^2}{d} \cdot \mathbb{E}\left[(y_{i_t'} - y_{i_t'}^t)^2 | \mathcal{F}_t\right]$$
$$= \frac{R^2}{nd}\|y - y^t\|^2.$$

And we can bound other terms in (20) similarly, then we obtain (18). The proof for variance bound (19) is analogous. $\square$

The following lemma is the key to prove the convergence of SPD1-VR:

**Lemma 8.** *Assume $g(x)$ is $\mu$-strongly convex, and all $\phi_i$ is $(1/\gamma)$-smooth, then by conditioning on $\mathcal{F}_t$, it holds that*

$$\frac{1}{2\eta}\mathbb{E}\left[\left(1 - \frac{\eta\mu}{4d} + \frac{12R'^2\eta\tau}{nd}\right)\|x^t - x\|^2 - \|x^{t+1} - x\|^2 - \left(\frac{1}{2} - \frac{8\eta\tau R'^2}{n}\right)\|x^t - \bar{x}^t\|^2\Big|\mathcal{F}_t\right]$$

$$+ \frac{1}{2\tau}\mathbb{E}\left[\left(1 - \frac{\tau\gamma}{4nd} + \frac{12R^2\eta\tau}{nd}\right)\|y^t - y\|^2 - \|y^{t+1} - y\|^2 - \left(\frac{1}{2} - \frac{8\eta\tau R^2}{d}\right)\|y^t - \bar{y}^t\|^2\Big|\mathcal{F}_t\right]$$

$$\geq \frac{1}{d}\cdot\mathbb{E}\left[F(x'^t, y) - F(x, y'^t)|\mathcal{F}_t\right] - \frac{4\eta R^2}{nd}\|\tilde{y}^k - y\|^2 - \frac{4\tau R'^2}{nd}\|\tilde{x}^k - x\|^2,$$

*for any $x \in X$ and $y \in Y$.*

*Proof.* First, by $2a\cdot b = (a+b)^2 - a^2 - b^2$, we have the following two inequalities:

$$2(x^t - \bar{x}^t)^\top(\bar{x}^t - x^{t+1}) = \|x^t - x^{t+1}\|^2 - \|x^{t+1} - \bar{x}^t\|^2 - \|x^t - \bar{x}^t\|^2$$

and

$$2(x^t - x^{t+1})^\top(x^{t+1} - x) = \|x^t - x\|^2 - \|x^{t+1} - x\|^2 - \|x^t - x^{t+1}\|^2$$

for any $x \in X$. By adding these two inequality together, we have

$$\|x^t - x\|^2 - \|x^{t+1} - x\|^2 - \|x^{t+1} - \bar{x}^t\|^2 - \|x^t - \bar{x}^t\|^2$$
$$= 2(x^t - \bar{x}^t)^\top(\bar{x}^t - x^{t+1}) + 2(x^t - x^{t+1})^\top(x^{t+1} - x)$$
$$= 2(x^t_{j_t} - \bar{x}^t_{j_t})\cdot(\bar{x}^t_{j_t} - x^{t+1}_{j_t}) + 2(x^t_{j_t} - x^{t+1}_{j_t})\cdot(x^{t+1}_{j_t} - x_{j_t}), \tag{21}$$

where the last line is because $x^t$ and $\bar{x}^t$ only differs in coordinate $j_t$, and similarly for $x^t$ and $x^{t+1}$. According to the updating rules

$$\bar{x}^t_{j_t} = \mathrm{prox}_{\eta g_{j_t}}\left(x^t_{j_t} - \eta\nabla^t_{x,j_t}\right),$$
$$x^{t+1}_{j_t} = \mathrm{prox}_{\eta g_{j_t}}\left(x^t_{j_t} - \eta\nabla'^t_{x,j_t}\right),$$

and the optimality condition of the proximal mapping subproblem, there must exist $s \in \partial g_{j_t}(\bar{x}^t_{j_t}) + \partial\mathbb{1}_{X_{j_t}}(\bar{x}^t_{j_t})$ and $s' \in \partial g_{j_t}(x^{t+1}_{j_t}) + \partial\mathbb{1}_{X_{j_t}}(x^{t+1}_{j_t})$ such that

$$\bar{x}^t_{j_t} = x^t_{j_t} - \eta\nabla^t_{x,j_t} - \eta s,$$
$$x^{t+1}_{j_t} = x^t_{j_t} - \eta\nabla'^t_{x,j_t} - \eta s',$$

where $\mathbb{1}_Z(z)$ is the indicator function of convex set $Z$, which takes value $0$ when $z \in Z$, while $\mathbb{1}_Z(z) = +\infty$ when $z \notin Z$. Combine these facts into (21), one can obtain:

$$\|x^t - x\|^2 - \|x^{t+1} - x\|^2 - \|x^{t+1} - \bar{x}^t\|^2 - \|x^t - \bar{x}^t\|^2$$
$$= 2\eta s\cdot(\bar{x}^t_{j_t} - x^{t+1}_{j_t}) + 2\eta s'\cdot(x^{t+1}_{j_t} - x_{j_t}) + 2\eta\nabla^t_{x,j_t}\cdot(\bar{x}^t_{j_t} - x^{t+1}_{j_t}) + 2\eta\nabla'^t_{x,j_t}\cdot(x^{t+1}_{j_t} - x_{j_t}).$$

Because of the separable assumption (2), $\mu$-strongly convex of $g(x)$ implies $\mu$-strongly convex of every component function $g_j(x_j)$. Now we apply the convexity of indicator function $\mathbb{1}_{X_{j_t}}(x_{j_t})$ and the strong convexity of $g_{j_t}(x_{j_t})$, and further observe that all of $\bar{x}^t, x^{t+1}$ and $x$ are always feasible, which mean $\mathbb{1}_{X_{j_t}}(\bar{x}^t_{j_t}) = \mathbb{1}_{X_{j_t}}(x^{t+1}_{j_t}) = \mathbb{1}_{X_{j_t}}(x_{j_t}) = 0$, and have

$$\|x^t - x\|^2 - \|x^{t+1} - x\|^2 - \|x^{t+1} - \bar{x}^t\|^2 - \|x^t - \bar{x}^t\|^2$$
$$\geq 2\eta\left[g_{j_t}(\bar{x}^t_{j_t}) - g_{j_t}(x^{t+1}_{j_t}) + \frac{\mu}{2}\left(\bar{x}^t_{j_t} - x^{t+1}_{j_t}\right)^2\right] + 2\eta\left[g_{j_t}(x^{t+1}_{j_t}) - g_{j_t}(x_{j_t}) + \frac{\mu}{2}\left(x^{t+1}_{j_t} - x_{j_t}\right)^2\right]$$
$$\quad + 2\eta\nabla^t_{x,j_t}\cdot(\bar{x}^t_{j_t} - x^{t+1}_{j_t}) + 2\eta\nabla'^t_{x,j_t}\cdot(x^{t+1}_{j_t} - x_{j_t})$$
$$= 2\eta\left[g_{j_t}(\bar{x}^t_{j_t}) - g_{j_t}(x_{j_t}) + \frac{\mu}{2}\left(\bar{x}^t_{j_t} - x^{t+1}_{j_t}\right)^2 + \frac{\mu}{2}\left(x^{t+1}_{j_t} - x_{j_t}\right)^2\right]$$
$$\quad + 2\eta\nabla^t_{x,j_t}\cdot(\bar{x}^t_{j_t} - x^{t+1}_{j_t}) + 2\eta\nabla'^t_{x,j_t}\cdot(x^{t+1}_{j_t} - x_{j_t})$$
$$\geq 2\eta\left[g_{j_t}(\bar{x}^t_{j_t}) - g_{j_t}(x_{j_t}) + \frac{\mu}{4}\left(\bar{x}^t_{j_t} - x_{j_t}\right)^2\right] + 2\eta\nabla^t_{x,j_t}\cdot(\bar{x}^t_{j_t} - x^{t+1}_{j_t}) + 2\eta\nabla'^t_{x,j_t}\cdot(x^{t+1}_{j_t} - x_{j_t}),$$

where inequality $a^2 + b^2 \geq (1/2)(a+b)^2$ is used in the second inequality. When $\eta\mu \leq 1$, it holds that

$$\frac{1}{2}\|x^t - \bar{x}^t\|^2 + \frac{\eta\mu}{2}\left(\bar{x}_{j_t}^t - x_{j_t}\right)^2 \geq \frac{1}{2}(x_{j_t}^t - \bar{x}_{j_t}^t)^2 + \frac{\eta\mu}{2}\left(\bar{x}_{j_t}^t - x_{j_t}\right)^2 \geq \frac{\eta\mu}{4}\left(x_{j_t}^t - x_{j_t}\right)^2.$$

As a result, we have

$$\|x^t - x\|^2 - \frac{\eta\mu}{4}\left(x_{j_t}^t - x_{j_t}\right)^2 - \|x^{t+1} - x\|^2 - \|x^{t+1} - \bar{x}^t\|^2 - \frac{1}{2}\|x^t - \bar{x}^t\|^2$$

$$\geq 2\eta\left[g_{j_t}(\bar{x}_{j_t}^t) - g_{j_t}(x_{j_t})\right] + 2\eta\nabla_{x,j_t}^t \cdot (\bar{x}_{j_t}^t - x_{j_t}^{t+1}) + 2\eta\nabla_{x,j_t}'^t \cdot (x_{j_t}^{t+1} - x_{j_t})$$

$$= 2\eta\left[g_{j_t}(\bar{x}_{j_t}^t) - g_{j_t}(x_{j_t})\right] + 2\eta\nabla_{x,j_t}'^t \cdot (\bar{x}_{j_t}^t - x_{j_t}) + 2\eta\left(\nabla_{x,j_t}^t - \nabla_{x,j_t}'^t\right) \cdot (\bar{x}_{j_t}^t - x_{j_t}^{t+1})$$

$$\geq 2\eta\left[g_{j_t}(\bar{x}_{j_t}^t) - g_{j_t}(x_{j_t})\right] + 2\eta\nabla_{x,j_t}'^t \cdot (\bar{x}_{j_t}^t - x_{j_t}) - \eta^2\left(\nabla_{x,j_t}^t - \nabla_{x,j_t}'^t\right)^2 - (\bar{x}_{j_t}^t - x_{j_t}^{t+1})^2$$

$$\geq 2\eta\left[g_{j_t}(\bar{x}_{j_t}^t) - g_{j_t}(x_{j_t})\right] + 2\eta\nabla_{x,j_t}'^t \cdot (\bar{x}_{j_t}^t - x_{j_t}) - \eta^2\left(\nabla_{x,j_t}^t - \nabla_{x,j_t}'^t\right)^2 - \|\bar{x}^t - x^{t+1}\|^2,$$

where Cauchy-Schwarz inequality is used in the second inequality. After cancelling term $\|x^{t+1} - \bar{x}^t\|^2$ on both sides, we finally obtain

$$\|x^t - x\|^2 - \frac{\eta\mu}{4}\left(x_{j_t}^t - x_{j_t}\right)^2 - \|x^{t+1} - x\|^2 - \frac{1}{2}\|x^t - \bar{x}^t\|^2$$

$$\geq 2\eta\left[g_{j_t}(\bar{x}_{j_t}^t) - g_{j_t}(x_{j_t})\right] + 2\eta\nabla_{x,j_t}'^t \cdot (\bar{x}_{j_t}^t - x_{j_t}) - \eta^2\left(\nabla_{x,j_t}'^t - \nabla_{x,j_t}^t\right)^2$$

$$= 2\eta\left[g_{j_t}(\bar{x}_{j_t}^t) - g_{j_t}(x_{j_t})\right] + 2\eta a_{i_t j_t}\bar{y}_{i_t}^t \cdot (\bar{x}_{j_t}^t - x_{j_t}) - 2\eta\left(a_{i_t j_t}\bar{y}_{i_t}^t - \nabla_{x,j_t}'^t\right) \cdot (\bar{x}_{j_t}^t - x_{j_t}) - \eta^2\left(\nabla_{x,j_t}'^t - \nabla_{x,j_t}^t\right)^2$$

$$= 2\eta\left[g_{j_t}(x_{j_t}'^t) - g_{j_t}(x_{j_t})\right] + 2\eta a_{i_t j_t}y_{i_t}'^t \cdot (x_{j_t}'^t - x_{j_t}) - 2\eta\left(a_{i_t j_t}y_{i_t}'^t - \nabla_{x,j_t}'^t\right) \cdot (x_{j_t}'^t - x_{j_t}) - \eta^2\left(\nabla_{x,j_t}'^t - \nabla_{x,j_t}^t\right)^2,$$

where in the last line we replaced $\bar{x}_{j_t}^t$ by $x_{j_t}'^t$, and $\bar{y}_{i_t}^t$ by $y_{i_t}'^t$.

Similarly, we can derive an analogous bound for dual variable $y$:

$$\|y^t - x\|^2 - \frac{\tau\gamma}{4d}\left(y_{i_t}^t - y_{i_t}\right)^2 - \|y^{t+1} - y\|^2 - \frac{1}{2}\|y^t - \bar{y}^t\|^2$$

$$= \frac{2\tau}{d}\left[\phi_{i_t}^*(y_{i_t}'^t) - \phi_{i_t}^*(y_{i_t})\right] + 2\tau a_{i_t j_t}x_{j_t}'^t \cdot (y_{i_t} - y_{i_t}'^t) - 2\tau\left(a_{i_t j_t}x_{j_t}'^t - \nabla_{y,i_t}'^t\right) \cdot (y_{i_t} - y_{i_t}'^t) - \tau^2\left(\nabla_{y,i_t}'^t - \nabla_{y,i_t}^t\right)^2.$$

After diving them by $2\eta$ and $2\tau$ respectively, we add the above two inequalities together:

$$\frac{1}{2\eta}\left[\|x^t - x\|^2 - \frac{\eta\mu}{4}\left(x_{j_t}^t - x_{j_t}\right)^2 - \|x^{t+1} - x\|^2 - \frac{1}{2}\|x^t - \bar{x}^t\|^2\right]$$

$$+ \frac{1}{2\tau}\left[\|y^t - x\|^2 - \frac{\tau\gamma}{4d}\left(y_{i_t}^t - y_{i_t}\right)^2 - \|y^{t+1} - y\|^2 - \frac{1}{2}\|y^t - \bar{y}^t\|^2\right]$$

$$= g_{j_t}(x_{j_t}'^t) - g_{j_t}(x_{j_t}) - a_{i_t j_t}y_{i_t}'^t \cdot x_{j_t} + a_{i_t j_t}x_{j_t}'^t \cdot y_{i_t} + \frac{1}{d}\left[\phi_{i_t}^*(y_{i_t}'^t) - \phi_{i_t}^*(y_{i_t})\right]$$

$$- \left(a_{i_t j_t}y_{i_t}'^t - \nabla_{x,j_t}'^t\right) \cdot (x_{j_t}'^t - x_{j_t}) - \left(a_{i_t j_t}x_{j_t}'^t - \nabla_{y,i_t}'^t\right) \cdot (y_{i_t} - y_{i_t}'^t)$$

$$- \frac{\eta}{2}\left(\nabla_{x,j_t}'^t - \nabla_{x,j_t}^t\right)^2 - \frac{\tau}{2}\left(\nabla_{y,i_t}'^t - \nabla_{y,i_t}^t\right)^2. \tag{22}$$

Now, we need to take conditional expectation of this inequality, by conditioning on $\mathcal{F}_t$. First, since both $x_j'^t$ and $y_i'^t$ are independent of both $i_t$ and $j_t$ for any $i \in [n]$ and $j \in [d]$. Thus,

$$\mathbb{E}\left[g_{j_t}(x_{j_t}'^t) - g_{j_t}(x_{j_t})|\mathcal{F}_t\right] = \frac{1}{d}\sum_{j=1}^d \mathbb{E}\left[g_j(x_j'^t) - g_j(x_j)|\mathcal{F}_t\right] = \frac{1}{d}\mathbb{E}\left[g(x'^t) - g(x)|\mathcal{F}_t\right],$$

$$\mathbb{E}\left[\phi_{i_t}^*(y_{i_t}'^t) - \phi_{i_t}^*(y_{i_t})|\mathcal{F}_t\right] = \frac{1}{n}\sum_{i=1}^n \mathbb{E}\left[\phi_i^*(y_i'^t) - \phi_i^*(y_i)|\mathcal{F}_t\right] = \frac{1}{n}\mathbb{E}\left[\phi^*(y'^t) - \phi^*(y)|\mathcal{F}_t\right],$$

$$\mathbb{E}\left[a_{i_t j_t}y_{i_t}'^t \cdot x_{j_t}|\mathcal{F}_t\right] = \frac{1}{nd}\sum_{i=1}^n\sum_{j=1}^d \mathbb{E}\left[a_{ij}x_j y_i'^t\right] = \frac{1}{nd}\mathbb{E}\left[y'^{t\top}Ax\right],$$

$$\mathbb{E}\left[a_{i_t j_t}x_{j_t}'^t \cdot y_{i_t}|\mathcal{F}_t\right] = \frac{1}{nd}\sum_{i=1}^n\sum_{j=1}^d \mathbb{E}\left[a_{ij}x_j'^t y_i\right] = \frac{1}{nd}\mathbb{E}\left[y^\top Ax'^t\right].$$

Second, by using the definition of $\nabla'^t_{x,j_t}$ and $\nabla'^t_{y,i_t}$, we have

$$\mathbb{E}\left[\left(a_{i_t j_t} y'^t_{i_t} - \nabla'^t_{x,j_t}\right) \cdot (x'^t_{j_t} - x_{j_t})|\mathcal{F}_t\right]$$
$$=\mathbb{E}\left[\left(a_{i_t,j_t} \tilde{y}^k_{i_t} - G^k_{x,j_t}\right) \cdot (x'^t_{j_t} - x_{j_t})|\mathcal{F}_t\right]$$
$$=\mathbb{E}\left[\mathbb{E}_{i_t}\left[a_{i_t,j_t} \tilde{y}^k_{i_t} - G^k_{x,j_t}\right] \cdot (x'^t_{j_t} - x_{j_t})|\mathcal{F}_t\right]$$
$$=\mathbb{E}\left[0 \cdot (x'^t_{j_t} - x_{j_t})|\mathcal{F}_t\right] = 0,$$

and similarly

$$\mathbb{E}\left[\left(a_{i_t j_t} x'^t_{j_t} - \nabla'^t_{y,i_t}\right) \cdot (y_{i_t} - y'^t_{i_t})|\mathcal{F}_t\right] = 0.$$

Finally, observe that

$$\mathbb{E}\left[\left(x^t_{j_t} - x_{j_t}\right)^2|\mathcal{F}_t\right] = \frac{1}{d}\|x^t - x\|^2,$$
$$\mathbb{E}\left[\left(y^t_{i_t} - y_{i_t}\right)^2|\mathcal{F}_t\right] = \frac{1}{n}\|y^t - y\|^2,$$

because both $x^t$ and $x$ is deterministic when conditioned $\mathcal{F}_t$. By putting all these facts back into (22), we can get:

$$\frac{1}{2\eta}\mathbb{E}\left[\left(1 - \frac{\eta\mu}{4d}\right)\|x^t - x\|^2 - \|x^{t+1} - x\|^2 - \frac{1}{2}\|x^t - \bar{x}^t\|^2\Big|\mathcal{F}_t\right]$$
$$+ \frac{1}{2\tau}\mathbb{E}\left[\left(1 - \frac{\tau\gamma}{4nd}\right)\|y^t - y\|^2 - \|y^{t+1} - y\|^2 - \frac{1}{2}\|y^t - \bar{y}^t\|^2\Big|\mathcal{F}_t\right]$$
$$\geq \frac{1}{d} \cdot \mathbb{E}\left[g(x'^t) - g(x) - \frac{1}{n}y'^{t\top} Ax + \frac{1}{n}y^\top Ax'^t + \frac{1}{n}\left[\phi^*(y'^t) - \phi^*(y)\right]\Big|\mathcal{F}_t\right]$$
$$- \frac{\eta}{2}\mathbb{E}\left[\left(\nabla'^t_{x,j_t} - \nabla^t_{x,j_t}\right)^2|\mathcal{F}_t\right] - \frac{\tau}{2}\mathbb{E}\left[\left(\nabla'^t_{y,i_t} - \nabla^t_{y,i_t}\right)^2|\mathcal{F}_t\right]$$
$$= \frac{1}{d} \cdot \mathbb{E}\left[F(x'^t, y) - F(x, y'^t)|\mathcal{F}_t\right] - \frac{\eta}{2}\mathbb{E}\left[\left(\nabla'^t_{x,j_t} - \nabla^t_{x,j_t}\right)^2|\mathcal{F}_t\right] - \frac{\tau}{2}\mathbb{E}\left[\left(\nabla'^t_{y,i_t} - \nabla^t_{y,i_t}\right)^2|\mathcal{F}_t\right].$$

Now, we apply Lemma 7 to bound the variance terms, and rearrange terms, then get

$$\frac{1}{2\eta}\mathbb{E}\left[\left(1 - \frac{\eta\mu}{4d} + \frac{12R'^2\eta\tau}{nd}\right)\|x^t - x\|^2 - \|x^{t+1} - x\|^2 - \left(\frac{1}{2} - \frac{8\eta\tau R'^2}{n}\right)\|x^t - \bar{x}^t\|^2\Big|\mathcal{F}_t\right]$$
$$+ \frac{1}{2\tau}\mathbb{E}\left[\left(1 - \frac{\tau\gamma}{4nd} + \frac{12R^2\eta\tau}{nd}\right)\|y^t - y\|^2 - \|y^{t+1} - y\|^2 - \left(\frac{1}{2} - \frac{8\eta\tau R^2}{d}\right)\|y^t - \bar{y}^t\|^2\Big|\mathcal{F}_t\right]$$
$$\geq \frac{1}{d} \cdot \mathbb{E}\left[F(x'^t, y) - F(x, y'^t)|\mathcal{F}_t\right] - \frac{4\eta R^2}{nd}\|\tilde{y}^k - y\|^2 - \frac{4\tau R'^2}{nd}\|\tilde{x}^k - x\|^2,$$

which is the desired result. $\qquad\square$

Now, we are ready to prove Theorem 3:

*Proof of Theorem 3.* First, we apply Lemma 8, and set $(x, y) = (x^*, y^*)$, we can get

$$\frac{1}{2\eta}\mathbb{E}\left[\left(1 - \frac{\eta\mu}{4d} + \frac{12R'^2\eta\tau}{nd}\right)\|x^t - x^*\|^2 - \|x^{t+1} - x^*\|^2 - \left(\frac{1}{2} - \frac{8\eta\tau R'^2}{n}\right)\|x^t - \bar{x}^t\|^2\Big|\mathcal{F}_t\right]$$
$$+ \frac{1}{2\tau}\mathbb{E}\left[\left(1 - \frac{\tau\gamma}{4nd} + \frac{12R^2\eta\tau}{nd}\right)\|y^t - y^*\|^2 - \|y^{t+1} - y^*\|^2 - \left(\frac{1}{2} - \frac{8\eta\tau R^2}{d}\right)\|y^t - \bar{y}^t\|^2\Big|\mathcal{F}_t\right]$$
$$\geq \frac{1}{d} \cdot \mathbb{E}\left[F(x'^t, y^*) - F(x^*, y'^t)|\mathcal{F}_t\right] - \frac{4\eta R^2}{nd}\|\tilde{y}^k - y^*\|^2 - \frac{4\tau R'^2}{nd}\|\tilde{x}^k - x^*\|^2$$
$$\geq -\frac{4\eta R^2}{nd}\|\tilde{y}^k - y^*\|^2 - \frac{4\tau R'^2}{nd}\|\tilde{x}^k - x^*\|^2 \tag{23}$$

where $F(x, y^*) - F(x^*, y) \geq 0$ for any $x \in X$ and $y \in Y$ is a property of the optimal solution $(x^*, y^*)$ of saddle point problems.

In the next, we will bound the coefficients in (23) respectively. Because of our choice of step sizes, we have

$$\eta \cdot \tau = \frac{\gamma}{128R^2} \min\left\{\frac{d\kappa}{n\kappa'}, 1\right\} \cdot \frac{n\mu}{128R'^2} \min\left\{\frac{n\kappa'}{d\kappa}, 1\right\} = \frac{n\gamma\mu}{2^{14}R^2R'^2} \min\left\{\frac{d\kappa}{n\kappa'}, \frac{n\kappa'}{d\kappa}\right\}. \quad (24)$$

Since $\min\{\alpha, \alpha^{-1}\} \leq 1$ for any $\alpha > 0$, this equality implies

$$\frac{1}{2} - \frac{8\eta\tau R^2}{d} \geq \frac{1}{2} - \frac{n\gamma\mu}{2^{11}dR'^2} = \frac{1}{2} - \frac{1}{2^{11}\kappa'} \geq 0 \quad \text{and} \quad \frac{1}{2} - \frac{8\eta\tau R'^2}{n} \geq \frac{1}{2} - \frac{1}{2^{11}\kappa} \geq 0,$$

where the definition of $\kappa$ and $\kappa'$ along with the assumptions $\kappa \geq 1$ and $\kappa' \geq 1$ are used. Besides, also from (24) we can know:

$$\frac{4\eta R^2}{nd} = \frac{\gamma\mu}{2^{12}dR'^2 \cdot \tau} \cdot \min\left\{\frac{d\kappa}{n\kappa'}, \frac{n\kappa'}{d\kappa}\right\} = \frac{1}{2^{12}n\kappa' \cdot \tau} \cdot \min\left\{\frac{d\kappa}{n\kappa'}, \frac{n\kappa'}{d\kappa}\right\} \leq \frac{1}{2^{12}\max\{n\kappa', d\kappa\} \cdot \tau}$$

$$= \frac{1}{2^{12}m \cdot \tau},$$

where we denote $m \triangleq \max\{n\kappa', d\kappa\}$ for simplicity, and similarly we can show that

$$\frac{4\tau R'^2}{nd} \leq \frac{1}{2^{12}m \cdot \eta}.$$

Furthermore, observe that

$$\begin{aligned}
\frac{\eta\mu}{4d} - \frac{12R'^2\eta\tau}{nd} &= \frac{\mu\eta}{4d}\left(1 - \frac{48R'^2\tau}{n\mu}\right) \\
&= \frac{\mu\eta}{4d}\left(1 - \frac{48}{128} \cdot \min\left\{\frac{n\kappa'}{d\kappa}, 1\right\}\right) \\
&\geq \frac{\eta\mu}{4d} \cdot \frac{5}{8} \\
&= \frac{5\mu\gamma}{2^{12}dR^2} \cdot \min\left\{\frac{d\kappa}{n\kappa'}, 1\right\} \\
&= \frac{5}{2^{12}d\kappa} \cdot \min\left\{\frac{d\kappa}{n\kappa'}, 1\right\} \\
&= \frac{5}{2^{12}m},
\end{aligned}$$

and

$$\frac{\tau\gamma}{4nd} - \frac{12R^2\eta\tau}{nd} \geq \frac{5}{2^{12}m}.$$

Combining all these facts into (23), and rearranging terms, we can get the recursive relationship:

$$\begin{aligned}
&\mathbb{E}\left[\frac{\|x^{t+1} - x^*\|^2}{\eta} + \frac{\|y^{t+1} - y^*\|^2}{\tau}\bigg|\mathcal{F}_t\right] \\
&\leq \left(1 - \frac{5}{2^{12}m}\right)\left[\frac{\|x^t - x^*\|^2}{\eta} + \frac{\|y^t - y^*\|^2}{\tau}\right] + \frac{1}{2^{11}m}\left[\frac{\|\tilde{x}^k - x^*\|^2}{\eta} + \frac{\|\tilde{y}^k - y^*\|^2}{\tau}\right] \\
&= \left(1 - \frac{5}{2^{12}m}\right)\left[\frac{\|x^t - x^*\|^2}{\eta} + \frac{\|y^t - y^*\|^2}{\tau}\right] + \frac{1}{2^{11}m}\Delta_k.
\end{aligned}$$

Now, we recursively apply this inequality, and use the law of total expectation, we can finally obtain:

$$
\begin{aligned}
&\mathbb{E}\left[\Delta_{k+1}\right] \\
=&\mathbb{E}\left[\frac{\|x^T - x^*\|^2}{\eta} + \frac{\|y^T - y^*\|^2}{\tau}\right] \\
\leq&\left(1 - \frac{5}{2^{12}m}\right)\mathbb{E}\left[\frac{\|x^{T-1} - x^*\|^2}{\eta} + \frac{\|y^{T-1} - y^*\|^2}{\tau}\right] + \frac{1}{2^{11}m}\cdot\Delta_k \\
\leq&\cdots \\
\leq&\left(1 - \frac{5}{2^{12}m}\right)^T\cdot\left[\frac{\|x^0 - x^*\|^2}{\eta} + \frac{\|y^0 - y^*\|^2}{\tau}\right] + \frac{1}{2^{11}m}\sum_{t=0}^{T-1}\left(1 - \frac{5}{2^{12}m}\right)^t\Delta_k \\
=&\left(1 - \frac{5}{2^{12}m}\right)^T\cdot\Delta_k + \frac{1}{2^{11}m}\cdot\frac{1 - [1 - 5/(2^{12}m)]^T}{1 - [1 - 5/(2^{12}m)]}\Delta_k \\
\leq&\left(1 - \frac{5}{2^{12}m}\right)^T\cdot\Delta_k + \frac{2}{5}\Delta_k.
\end{aligned}
$$

Therefore, to finish the proof, we only need to make sure $(1 - 5/(4096m))^T \leq 1/5$, which can be guaranteed if choosing some large enough $T \geq \Theta(m) = \Theta(\max\{d\kappa, n\kappa'\})$. $\qquad\square$

Figure 3: Numerical results on the problem of squared-hinge loss SVM. The $y$-axis is primal sub-optimality.

# 2 Extra Numerical Experiments

In this part, we will show some extra experiment results. The experiment setting is basically same to Section 5. The only difference is that we change the model to support vector machine (SVM) with squared-hinge loss, i.e.,

$$\phi_i(u) = \max\{0, 1 - b_i u\}^2$$

where $b_i \in \{\pm 1\}$ is the class label. Note that the corresponding conjugate function is

$$\phi_i^*(y) = \left\{ \begin{array}{ll} b_i y + \frac{y^2}{4}, & \text{if } b_i y \leq 0, \\ +\infty, & \text{if } b_i y > 0, \end{array} \right.$$

whose proximal mapping has simple closed-form solution, and does not need to be solved iteratively.

The results are shown in Figure 3. The performance of all methods are similar to that in Section 5.

Besides, we further report the running time of all methods in Table 2. We can observe that SPD1 and SPD1-VR takes longer time than SGD and SVRG for each pass of data. We think there are mainly two reasons which result in such phenomenon: 1) SPD1 and SPD1-VR involves much more loops, which may incur computation overhead (our codes are written in Julia), while the updates of SGD and SVRG can be implemented in vector forms, and the vector operations are conducted by some highly-optimized computation libraries like OpenBLAS; 2) sampling only one scalar from the data matrix each time is cache-unfriendly for computers. However, we believe these two issues can be solved. For example, the former issue can be tackled by using faster programming languages like C/C++. While for the latter one, one possible way to solve it is adopting mini-batch versions of SPD1 and SPD1-VR, which sample a batch of continuous coordinates instead of just one in each iteration.

Table 2: Running time required by different methods for one pass of data (in seconds)

| Methods | colon-cancer | gisette | rcv1.binary |
|---------|--------------|---------|-------------|
| SPD1 | 0.012 | 2.70 | 128.4 |
| SPD1-VR | 0.013 | 3.62 | 105.5 |
| SGD | 0.006 | 1.07 | 34.8 |
| SVRG | 0.006 | 1.37 | 45.1 |