[Reviews · NeurIPS 2018]

Reviewer 1



In this paper, the author has proposed doubly stochastic like primal-dual method to solve empirical risk minimization with linear predictors. As this method considers only one data point and one direction at a time, hence per-iteration complexity is of the constant order. The authors also propose a variance reduced optimization approach. The paper has been written clearly. The idea of using this kind of update has been around for some time. A minor concern is as following: I have not seen an application where it is really helpful because the SGD and coordinate descent are both cheap. Hence I would like to see the analysis of this algorithm in distributed environment. Also I am not sure right now that why do the bound in theorem 2 depends on the dimension d which makes this approach useless for all the kernel base algorithms. However coordinate descent or SGD kind of algorithm just run fine in kernel based learning algorithms also. Where does the analysis break ? Am I missing something ? I am keeping my score right now 6 under the assumption that I might have misunderstood something, however depending on the clarification from the authors, I am happy to change my score.

Reviewer 2



# Summary This paper proposes a new learning method for minimizing convex finite sums with strongly convex regularizer and linear predictor. One notable feature of the method is the per-iteration computational cost is O(1) while usual SGD needs O(d). The idea of the method is to take primal-dual reformulation and apply the doubly stochastic primal-dual method to the problem. A variance reduced variant of the method is also proposed by combining SVRG, naturally. For convex and strongly convex problems, this paper derives convergence rates of these two methods. Experimental results show the superior performance of the proposed method over competitors including notable methods such as SGD, SVRG, and SAGA. Especially, the method outperforms the other methods in high-dimensional settings. # Detailed comments The algorithm derivation and obtained algorithms seem to be reasonable and interesting. A convergence analysis for non-strongly convex problems is also good, which corresponds to the rate of SGD. However, there is much room for improvement of an analysis for strongly convex problems as mentioned in the paper. The paper argues about only the dependency on $T$, but another concern is the dependency on $d$ and $n$. In the case of PSGD, convergence rates are independent of $n$, but the proposed method is affected by both $d$ and $n$. For the SPD1-VR, it would be nice if the authors could investigate large stepsize is acceptable in the method to achieve a faster convergence rate. I think that the following paper is also useful for further develop this work. Nutini, Schmidt, Laradji, Friedlander, and Koepke. Coordinate Descent Convergence Faster with the Gauss-Southwell Rule than Random Selection. ICML, 2015 # After reading the author's response A new convergence rate is nice as a stochastic method. I vote for acceptance because my concern has been resolved.

Reviewer 3



******* After Rebuttal ***** The authors have addressed almost all my questions. I have decided not to increase my grade since my criticism on intuition & readability and time plots really depends on the quality of the re-write and the plots. In the rebuttal the authors have suggested that their method can be deduced using a saddle point formulation and they will include time plots, which I look forward to seeing. But still, it depends on the re-write. Furthermore on why I feel I cannot raise my grade, as pointed out by Reviewer 1, an application where data access is truly the bottleneck (such as a distributed setting) would be very welcome and would result in an excellent and well rounded paper i.e. good theoretical results, and a good practical result. ***************************** The papers introduces a method with O(1) data access and O(1) cost per iteration (under the assumption that the regularizer is separable) for solving the primal-dual formulation of empirical risk minimization for generalized linear models. This is the first time I’ve seen a O(1) per iteration cost, and thus I find the paper quite interesting in the regime where loading data is truly the bottleneck. On the downside, very little intuition or any form of a derivation of the algorithms is offered, making the paper only suitable for experts in convex optimization and proximal methods. Under the assumption that the loss functions are Lipschitz and convex, the authors establish the O(1/sqrt(t)) and O(1/t) convergence for their algorithm when the regularizer is convex and strongly convex, respectively. They also extend their algorithm to a variance reduced version with linear convergence (under the additional assumption that the loss functions are smooth), with a rate that is comparable to SVRG. These theoretical results are as strong as could be hoped for and, together with some limited experiments, is a substantial contribution, and for this reason I recommend accepting the paper. The reason I do not recommend a strong accept is due to a minor issues that I detail next. Extended reals and a bounded domain: For your proofs and calculations you assume g_j is a functional over a bounded domain which is why the solution to x’ = prox_{g_j} (x) depends on the indicator function as such x’ \in partial g_j(x’) +(x’-x) + \partial 1_{X_j}(x’) Thus g_j should be defined as g_j: X_j \mapsto \R \cup \{ \infty \}. This is only a minor detail, but important insofar that it occults the dependency on the bounded sets X_j and guarantees the feasibility of the iterates x^t. Without this clarification an inexperienced reader may not understand how feasibility is maintained. This raises the further question, what are these bounded domains in your experiments on logistic regression? And is the projection onto these bounded domains computed in practice, or is this only a technical requirement for the proofs? This should be clarified. Clarify assumptions earlier on: I believe the assumptions need to be brought forward. Since the major selling points of the algorithm is that each iteration costs only O(1), as stated in the abstract. But this only holds for separably regularizers where both the loss and regularizer are prox friendly. This would even fit and be appropriate in the abstract. Intuition & readability: Very little intuition of explanations are offered for Algorithm 1 and 2 aside from stating that Algorithm 1 is, in some sense, a randomized coordinate descent type method. But this is not clear, and without such clarifications I feel the paper is only accessible to experts. The paper would benefit from a derivation or better explanation of either algorithm. In particular, relating the development of your methods to the Primal Dual Hybrid Gradient (PDHG) of Chambolle and Pock [1] would help ground the reader. Furthermore it seems this paper [2] might be quite related since they do formally take a coordinate ascent step in a PDHG type method (though they do not use and stochastic gradient step). Note you still have 1/3 of a page to fill before reaching the 8 page maximum. Experiments: The experiments only report the efficiency of the algorithms in terms of data passes. I understand that in certain settings this may indeed be the dominant cost, and the experiments nicely illustrate how the new algorithms perform well in this setting since they only require O(1) data access per iteration. But including a metric of either time or approximate flops would be very informative. It is understandable that these new O(1) algorithms will perhaps not perform well in clock time on a single cpu, but it would enrich the understanding of the empirical performance of these new algorithms to have additional metrics. This can also be partly remedied by simulating a distributed setting, or implementing a parallel implementation. Another minor issue with the experiments is that the logistic loss is not truly prox friendly. Though, as pointed out by the authors, a prox step over a logistic loss can be efficiently computed with a Newton type method in one dimension, this is still not the same as an exact prox step, indeed you cannot perform an exact prox step using an iterative algorithm. Finally, I found the following minor mistakes: line 16: … independent of n line 20: … at a sublinear rate … line 193: To allow for greater reproducibility, the authors should explain how they tuned the stepsizes. A grid search over a certain grid? Supplementary material: line 20: There is a missing \partial’’ from s \in \partial g_j(x_j^{‘t}) + \partial ... Sentences ending with a comma: such as on line 28 and 32. [1] A first-order primal-dual algorithm for convex problems with applications to imaging, Antonin Chambolle, Thomas Pock [2] STOCHASTIC PRIMAL-DUAL HYBRID GRADIENT ALGORITHM WITH ARBITRARY SAMPLING AND IMAGING APPLICATIONS, Antonin Chambolle, Matthias Ehrhardt, Peter Richtarik and Carola-Bibiane Schönlieb. Justifying confidence score: I am an expert in stochastic algorithms for solving the convex empirical risk minimization problem. I am also very familiar with randomized coordinate type algorithms in this and other settings. This is why I am confident in my assessment.